# Meta Pruning via Graph Metanetworks : A Universal Meta Learning Framework for Network Pruning

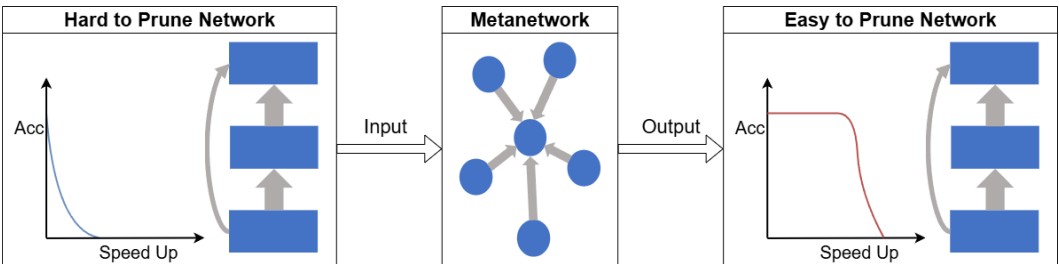

Figure 1: For a pruning criterion, we use a metanetwork to change a hard to prune network into another easy to prune network for better pruning.

## Abstract

We propose an entirely new meta-learning framework for network pruning. It is a general framework that can be theoretically applied to almost all types of networks with all kinds of pruning and has great generality and transferability. Experiments have shown that it can achieve outstanding results on many popular and representative pruning tasks (including both CNNs and Transformers). Unlike all prior works that either rely on fixed, hand-crafted criteria to prune in a coarse manner, or employ learning to prune ways that require special training during each pruning and lack generality. Our framework can learn complex pruning rules automatically via a neural network (metanetwork) and has great generality that can prune without any special training. More specifically, we introduce the newly developed idea of metanetwork from meta-learning into pruning. A metanetwork is a network that takes another network as input and produces a modified network as output. In this paper, we first establish a bijective mapping between neural networks and graphs, and then employ a graph neural network as our metanetwork. We train a metanetwork that learns the pruning strategy automatically and can transform a network that is hard to prune into another network that is much easier to prune. Once the metanetwork is trained, our pruning needs nothing more than a feedforward through the metanetwork and some standard finetuning to prune at state-of-the-art. Our code is available at `https://anonymous.4open.science/r/MetaPruning`.

## 1 Introduction

With the rapid advancement of deep learning (LeCun et al., 2015; Schmidhuber, 2015), neural networks have become increasingly powerful. However, this improved performance often comes with a significant increase in the number of parameters and computational cost (FLOPs). As a result, there is growing interest in methods to simplify these networks while preserving their performance. Pruning, which involves selectively removing certain parts of a neural network, has proven to be an effective approach (Cheng et al., 2024; He & Xiao, 2024; Reed, 1993).

A key idea of a large number of previous pruning works is to remove the unimportant components of a neural network. To achieve this goal, various criteria that measure the importance of components of

a neural network have been invented. Some criteria are based on norm or magnitude scores (Han et al., 2016; Li et al., 2017b; Sun et al., 2024; He et al., 2018a). Some are based on how much the performance changes when certain parts of the network are removed or preserved (You et al., 2019b; Ma et al., 2023; Frankle & Carbin, 2019; Ye et al., 2020). Some others defined more complex criteria such as saliency and sensitivity (LeCun et al., 1989; Lee et al., 2019; Zhao et al., 2019).

Once a pruning criterion is established, we can assess whether a model is easy or hard to prune based on that criterion. Technically, we consider a model easy to prune if we can prune a relatively large part of it with only little loss in accuracy. Otherwise, we consider it hard to prune.

Prior researchers have generally approached pruning improvement in two ways: either by refining the pruning criteria or by transforming the model to be easier to prune according to a given criterion. The first approach is more straightforward and has been extensively studied in prior work. In contrast, the second approach has received comparatively less attention. However, several studies adopting this latter perspective have achieved promising results. Notably, sparsity regularization methods (Wen et al., 2016; Fang et al., 2023; Wang et al., 2021) exemplify this approach by encouraging models to become sparser, thereby facilitating more effective pruning.

Our framework inherits the idea of the second way. A metanetwork is a network that takes another network as input and produces a modefied network as output. For any given pruning criterion, our framework aims at training a metanetwork that can transform a hard to prune network into another easier to prune network like figure 1.

Our contributions can be summarized as follows:

(1) Introduce the idea of metanetwork from meta-learning into pruning for the first time.

(2) Propose an entirely new universal pruning framework that is theoretically applicable to almost all types of networks with all kinds of pruning.

(3) Present concrete implementations for our theoretical framework and achieve state-of-the-art performance on various practical pruning tasks.

(4) Conduct further research on the flexibility and generality of our framework.

Our method is entirely new and fundamentally different from all prior works, see Appendix A for a comparison with prior works to better understand our novel contributions and advantages.

## 2 RELATED WORK

### 2.1 METANETWORKS

Using neural networks to process other neural networks has emerged as an intriguing research direction. Early studies demonstrated that neural networks can extract useful information directly from the weights of other networks (Unterthiner et al., 2020). We define a *metanetwork* as a neural network that takes another neural network as input and outputs either information about it or a modified network. Initial metanetworks were simple multilayer perceptrons (MLPs), while more recent designs incorporate stronger inductive biases by preserving symmetries (Godfrey et al., 2022) inherent in the input networks. Broadly, metanetwork architectures can be categorized into two perspectives. The *weight space view* applies specially designed MLPs directly on the model's weights (Zaheer et al., 2017; Navon et al., 2023; Zhou et al., 2023a; Tran et al., 2024b; Zhou et al., 2023b; 2024; Tran et al., 2024a). And the *graph view* transforms the input network into a graph and applies graph neural networks (GNNs) to it (Lim et al., 2024; Kofinas et al., 2024; Kalogeropoulos et al., 2024). In this work, we adopt the graph view to build our metanetwork, leveraging the natural correspondence between graph nodes and neurons in a network layer, as well as the symmetries they exhibit (Maron et al., 2019). Given the maturity and effectiveness of GNNs, the graph metanetwork approach offers both elegance and practicality.

### 2.2 GRAPH NEURAL NETWORKS

Graph neural networks (GNNs) are a class of neural networks specifically designed to operate on graph-structured data, where graphs consist of nodes and edges with associated features (Wu et al., 2021; Scarselli et al., 2009; Kipf & Welling, 2017). GNNs have gained significant attention in recent

years, with frameworks such as the message passing neural network (MPNN) (Gilmer et al., 2017) proving to be highly effective. In this paper, we employ the message passing framework, specifically using Principal Neighborhood Aggregation (PNA) (Corso et al., 2020) as the backbone architecture for our metanetwork.

### 2.3 SPARSITY REGULARIZATION BASED PRUNING

Networks that are inherently sparse tend to be easier to prune effectively. Prior research has explored various regularization techniques to encourage sparsity in neural networks to facilitate pruning. This idea of sparsity regularization has been widely adopted across many works (Wen et al., 2016; Gordon et al., 2018; Huang & Wang, 2018; Lin et al., 2020c; He et al., 2017; Lin et al., 2019; Xia et al., 2022; 2024; Fang et al., 2023; Wang et al., 2021). Our method inherits this principle to some extent, as our metanetwork can be viewed as transforming the original network into a sparser version for better pruning.

### 2.4 LEARNING TO PRUNE & META PRUNING

Given the complexity of pruning, several previous works have leveraged neural networks (Dery et al., 2024; He et al., 2017; Wu et al., 2024; Chen et al., 2023), reinforcement learning (Rao et al., 2019; Yu et al., 2022; He et al., 2018b), and other techniques to automatically learn pruning strategies. Viewing pruning as a learning problem naturally leads to the idea of meta-learning, which learns "how to learn" (Hospedales et al., 2022). Some prior works have specifically applied meta-learning to pruning (Liu et al., 2019; Li et al., 2020). Our work also draws idea from meta-learning, but it learns in a way entirely different from all prior works.

## 3 A UNIVERSAL META-LEARNING FRAMEWORK

Our framework can be summarized in one single sentence (Figure 1):

> **For a pruning criterion, we use a metanetwork to change a hard to prune network into another easy to prune network for better pruning.**

**Pruning criterion:** A pruning criterion measures the importance of specific components of a neural network. Typical examples include the $\ell_1$ norm and $\ell_2$ norm of weight vectors. During pruning, we compute an importance score for each prunable component according to the criterion, and remove components in ascending order of their scores.

**Easy or hard to prune:** For a fixed pruning criterion, if we gradually prune the network and the accuracy drops quickly, we refer to it as *hard to prune*. Conversely, if the accuracy decreases slowly with progressive pruning, we call it *easy to prune*.

**Metanetwork:** A metanetwork is a special type of neural network whose inputs and outputs are both neural networks. In contrast, standard neural networks typically take structured data such as images (CNNs), sentences (Transformers), or graphs (GNNs) as input. A metanetwork, however, takes an existing network as input and outputs another network, hence the term *meta*—indicating that it operates on neural networks themselves.

## 4 A SPECIFIC IMPLEMENTATION BASED ON GRAPH METANETWORKS

The proposed framework is conceptually simple and straightforward. However, its practical implementation requires addressing several technical challenges:

(1) How to build the metanetwork ?

(2) How to train the metanetwork ?

(3) How to design the pruning criterion ?

We present our implementations step by step, dedicating one subsection to each challenge. The final subsection offers an overall summary of our method and a deeper understanding of it from a more holistic perspective.

### 4.1 METANETWORK DESIGN

Drawing ideas from recent researches (section 2.1), we use a graph neural network (GNN) as our metanetwork. Please make sure you are familiar with the basic concepts of GNN before continuing. Our design does not cover any complex GNN theory, you only need to know what is a graph and the basic message passing algorithm.

**Conversion between networks and graphs:** To apply a metanetwork (GNN) on networks, we first need to establish a conversion between networks and graphs. Prior works such as Kofinas et al. (2024); Lim et al. (2024) have managed to change networks of almost all architectures (CNN, Transformer, RNN, etc.) into graphs.

Here we show how we establish conversion between a ResNet (He et al., 2016) and a graph in our experiments as an example. This inherits the idea of Kofinas et al. (2024); Lim et al. (2024) but is quite different in implementation details. To convert a network into a graph, we establish the correspondence between the components of a neural network and elements of a graph as follows:

(1) **Node:** Each neuron in fully connected layers or channel in convolutional layers in the network is represented as a node in the graph. (2) **Edge:** An edge exists between two nodes if there is a direct connection between them in the original network. This includes fully connected layers, convolutional layers, and residual connections. (3) **Node Features:** Node features comprise parameters associated with neurons, here we use the weight, bias, running mean, running var

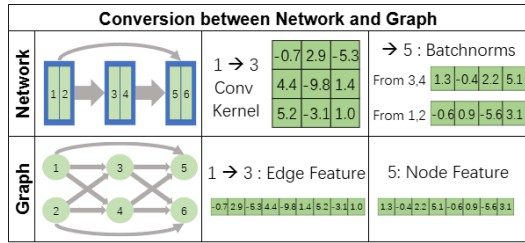

Figure 2: **Conversion between networks and graphs:** We visualize a two layer toy CNN as example. Our graph contains 6 nodes corresponding with 6 neurons in the network. For neurons connected via convolutional channel like (1,3) and connected by residual connection like (1,5), we add an edge between their corresponding nodes. Then we generate node and edge features as shown in the figure. For some special cases, we use default values to replace non-existent values. For example, for node 3 who doesn't have a residual connection, we set the last 4 number of its node feature as [1, 0, 0, 1]. This means we treat it as with previous residual batchnorm of weight 1, bias 0, running mean 0, running var 1, performing the same as with no batchnorm.

of batchnorm (Ioffe & Szegedy, 2015), including both batchnorms from previous adjacent layers and previous skip connection layers (if exist). (4) **Edge Features:** Edge features encode the weights of connections between nodes. We treat linear connections and residual connections as special cases of convolutional connections with a kernel size of 1. For a $k \times k$ convolution between two channels, we flatten it into a $k^2$-dim edge feature vector.

Since graph neural networks here require all edge features to have the same dimension, we adopt one of two strategies to standardize them: (1) Pad all convolutional kernels to the same size (e.g., the maximum kernel size in the network) (2) Flatten convolutional kernels of varying sizes, then apply learned linear transformations to project them into the same size.

During conversion, we simply ignored other components such as pooling layers. Although we have methods to explicitly convert all of them, our experiments show that omitting them already yields outstanding results. Therefore, it is acceptable to ignore them in practice. We also didn't use any other techniques like positional embedding.

By following these rules, we can generate a graph that corresponds to a given neural network. Conversely, the inverse transformation can be applied to convert a graph back into an equivalent network. In essence, we establish an equivalent conversion between networks and graphs. For a more intuitive understanding, see Figure 2.

**Metanetwork architecture:** After establishing the conversion between networks and graphs, we can convert the input network into a graph, use metanetwork(GNN) to transform the graph, and finally convert the output graph back into a new network. Our metanetwork must be powerful enough to learn the transforming rules for better pruning. We build our metanetwork based on the message passing framework (Gilmer et al., 2017) and PNA (Corso et al., 2020) architecture. A brief overview of our architecture is provided below, the full description is available in Appendix B.1.

We use $v_i$ to represent node $i$'s feature vector, and $e_{ij}$ to represent feature vector of the edge between $i$ and $j$. We consider undirected edges, i.e., $e_{ij}$ is the same as $e_{ji}$. Our metanetwork is as follows.

First, all input node and edge features are encoded into the same hidden dimension.

$$v \leftarrow \text{MLP}_{NodeEnc}(v_{in}), \tag{1}$$
$$e \leftarrow \text{MLP}_{EdgeEnc}(e_{in}). \tag{2}$$

Then they will pass through several message passing layers. In each layer, we update node and edge features using information from adjacent nodes and edges.

$$v_i \leftarrow f_v(v_i, v_j((i,j) \in E), e_{ij}((i,j) \in E)) \tag{3}$$
$$e_{ij} \leftarrow f_e(v_i, v_j, e_{ij}) \tag{4}$$

Here $f_v$ and $f_e$ are fixed calculating processes.

Finally, we use a decoder to recover node and edge features to their original dimensions:

$$v_{pred} \leftarrow \text{MLP}_{NodeDec}(v), \tag{5}$$
$$e_{pred} \leftarrow \text{MLP}_{EdgeDec}(e). \tag{6}$$

We multiply predictions by a residual coefficient and add them to the original inputs as final outputs:

$$v_{out} = \alpha \cdot v_{pred} + v_{in}, \tag{7}$$
$$e_{out} = \beta \cdot e_{pred} + e_{in}, \tag{8}$$

where $\alpha$ and $\beta$ in practice are set to a small real numbers like $0.01$. This design enables the metanetwork to learn only the delta weights on top of the original weights, instead of a complete reweighting.

## 4.2 META-TRAINING

Meta-training is the process we train our metanetwork (Figure 3).

**Data preparation:** Meta-training data is comprised of two parts. The first is a set of neural networks to prune, which serve as the origin network in meta-training. We call them *data models*. The number of data models can be arbitrary small numbers like 1, 2, or 8, because our implementations aren't sensitive to it (See more explanations in Section 4.4).The second is the traditional training datasets, CIFAR-10 for example, used to calculate the accuracy loss.

**One training iteration:** We select one model from data models as the origin network. It is converted into its graph representation, passed through the metanetwork to get a new graph and converted back into a new network. We calculate accuracy loss and sparsity loss on this new network, and backpropagated the gradients to update the metanetwork.

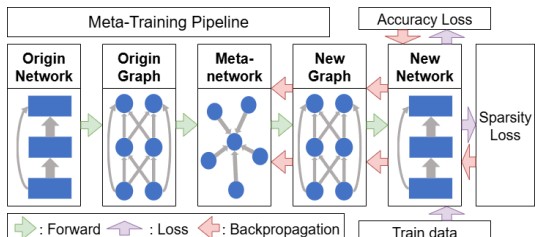

Figure 3: **Meta-Training Pipeline:** During each iteration, the origin network is converted into the origin graph, fed through the metanetwork to get a new graph and finally converted back into a new network. We calculate accuracy loss and sparsitly loss on the new network, then backpropagated the gradients to update the metanetwork.

Two types of losses are used during meta-training.**(1) Accuracy Loss**: We feed the training data (e.g., Train set of CIFAR-10. Note that test set is not used here) into the new network and compute the cross-entropy loss based on the output predictions, ensuring the metanetwork doesn't excessively disturbing the effective parts of the origin network. **(2) Sparsity Loss**: We calculate a regularization term on the new network to encourage it to be sparse, making it easier to prune. It is related to the design of pruning criterion (section 4.3)

### 4.3 PRUNING CRITERION

A valid pruning criterion consists of two components:**(1) An importance score function:** During pruning, we calculate the scores of all prunable components of a network, and prune them in ascending order of their scores.**(2) A sparsity loss:** It is a kind of loss that is calculated on the parameters of the network. Optimizing the network using this loss can make it easier to prune.

We perform most of our experiments by default using structural pruning, but also include a small number of experiments with unstructured and N:M sparsity pruning to demonstrate that our method can be applied to all kinds of pruning. In unstructured pruning and N:M sparisity pruning, we simply use the naive $l_1$ norm as both our importance score function and sparsity loss.

In structural pruning, we draw ideas from prior sparsity regularization based pruning methods(section 2.3), and design our pruning criterion based on them (especially Fang et al. (2023)). We will give a brief introduction below. For a rigorous mathematical definition and more details, please refer to appendix B.2.

A variant of the classical $\ell_2$ norm, the *group norm* is used as our importance score function. It inherits the idea of using $\ell_2$ norm to calculate a score, but calculate it on a pruning group level. Since structural pruning is performed on groups, traditional $l_2$ norm that treats nodes and edges in isolation is no longer viable. Group $l_2$ norm that takes a whole group into consideration has shown to be more effective. We calculate the sparisty loss by multiply our group norm scores with some coefficients.

The choice of pruning criteria is highly flexible (see in our later experiments, section 6.1), which further demonstrates the generality of our framework. We perform most of our experiments by default using structural pruning, but also include a small number of experiments with unstructured and N:M sparsity pruning to demonstrate that our method can be applied to all kinds of pruning.

### 4.4 A HOLISTIC PERSPECTIVE

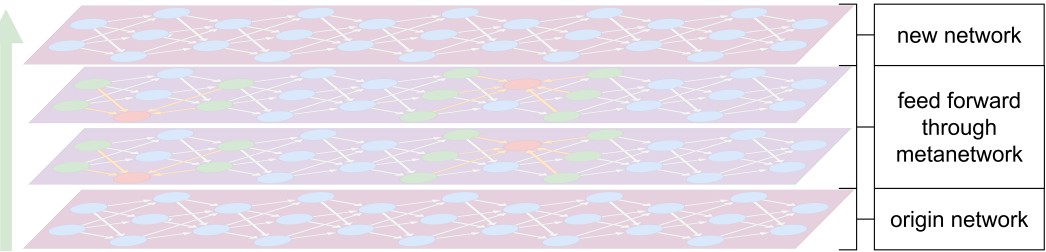

Figure 4: **Feed forward through metanetwork:** when a network is fed forward through the metanetwork, every parameter in it gathers information from neighbour parameters and architectures.

Intuitively, when a network is fed forward through the metanetwork, every parameter in it gathers information from neighbour parameters and architectures (Figure 4). The metanetwork automatically learns how to process the information from neighbours and change the weights of the network based on that for better pruning. And this has several important properties:

**Generality:** Our metanetwork is a really small GNN compared to the origin network, which means each parameter can only see information from a limited local region. Due to the nature of GNN, parameters at any position in the network—whether at the beginning, middle, or end—must process information from neighbours in the same way. As a result, the metanetwork must learn a general strategy to adjust parameters based on neighbour information for better pruning. This explains why, in all our experiments, the pruning models and meta-training data models are completely different, but the performance is as good as the same. This also explains why our implementations are not sensitive to the number of data models. Even a single data model can be viewed as a collection of numerous data points sufficient to train the metanetwork. So there is no overfitting at all, and whether we are using 1, 2 or 8 data models during meta-training, the results remain the same.

**Natural transferability:** Just as a GNN can be applied to graphs of varying sizes without modification, our metanetwork can be directly applied to networks of different scales. This property gives it natural transferability, as demonstrated in our experiments (section 6.3) where it successfully transfers across related datasets and network architectures.

**Universally applicability:** Theoretically, our implementations can be applied to almost all types of networks with all kinds of pruning. Almost all types of networks can be converted into graphs (Kofinas et al., 2024; Lim et al., 2024), and the designs of the GNN and the pruning criteria are also broadly compatible. This makes our framework a universally applicable solution for network pruning.

# 5 EXPERIMENTS ON CLASSICAL CNN PRUNING TASKS

## 5.1 PRELIMINARIES

$$\underbrace{\text{Initial Pruning (Finetuning)}}_{\text{Optional}} \rightarrow \underbrace{\text{Metanetwork (Finetuning)} \rightarrow \text{Pruning (Finetuning)}}_{\text{Necessary}} \quad (9)$$

Definitions of the terms *Speed Up* and *Acc vs. Speed Up Curve* are in Appendix C.1. The pruning pipeline is described as equation 9, and full details are provided in Appendix C.2.

Unlike prior learning to prune methods that require special training during each pruning, our pruning pipeline only requires a feed forward through metanetwork and standard finetuning, which is simple, general and effective. Once we have a metanetwork, we can prune as many networks as we want. All pruning models are completely different from meta-training data models. This demonstrates our metanetwork naturally has great transferability and no prior work has done something like this before. See Appendix A for more comparisons between our work and prior ones (ideas, costs etc.).

## 5.2 CLASSICAL CNN PRUNING TASKS

We carry out our experiments on three most classical, popular, and representative image recognition tasks, including pruning ResNet56 (He et al., 2016) on CIFAR10 (Krizhevsky & Hinton, 2009), VGG19 (Simonyan & Zisserman, 2015) on CIFAR100 (Krizhevsky & Hinton, 2009) and ResNet50 (He et al., 2016) on ImageNet (Deng et al., 2009). See appendix D.1 for more general setups.

Our method achieves outstanding results on all 3 tasks and is better than almost all prior works (Table 1). See full results compared with prior works in Table 8(ResNet56 on CIFAR10), Table 11(VGG19 on CIFAR100), Table 14(ResNet50 on ImageNet). See Appendix D.2 D.3 D.4 for implementation details on each tasks.

Table 1: **Results for 3 classical CNN pruning tasks** including pruning (1) ResNet56 on CIFAR10, (2) VGG19 on CIFAR100 (3) ResNet50 on ImageNet. For full results compared with prior works, see (1) Table 8, (2) Table 11, (3) Table 14.

| Task | Base Top-1(Top-5) | Pruned Top-1($\Delta$) | Pruned Top-5($\Delta$) | Pruned FLOPs |
|------|-------------------|------------------------|------------------------|--------------|
| (1)  | 93.51%            | 93.64%(+0.13%)         | —                      | 65.6%        |
| (2)  | 73.65%            | 69.75%(-3.90%)         | —                      | 88.83%       |
| (3)  | 76.14%(93.11%)    | 76.13%(-0.01%)         | 92.78%(-0.33%)         | 57.2%        |

## 5.3 ABLATION STUDY OF GENERAL BEHAVIORAL TENDENCIES

**How metanetwork works:** We visualized the "Acc vs. Speed Up" curve of both origin network and network after feedforward through metanetwork and finetuning (Figure 5). They show that as the pruning speed up increases, the accuracy drops at a significantly slower rate after applying the metanetwork and finetuning.

**Trade-off between accuracy and speed up:** During meta-training: as the number of training epochs increases, the metanetwork's ability to make the network easier to prune becomes stronger. However, its ability to maintain the accuracy becomes weaker. As shown in Figure 6a, with more training epochs, the "Acc VS. Speed Up" curve shifts downward, and the flat portion of the curve becomes longer. **This characteristic allows us to adaptively meet different pruning requirements.** If a higher level of pruning is desired and a moderate drop in accuracy is acceptable, a metanetwork trained for more epochs would be preferable. Conversely, if preserving accuracy is more important, a metanetwork trained for fewer epochs would be a better choice (Figure 6b).

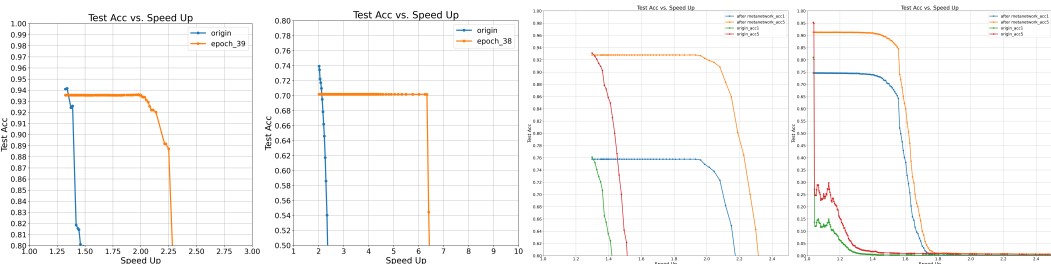

(a) ResNet56 on CIFAR10  (b) VGG19 on CIFAR100  (c) ResNet50 on ImageNet      (d) ViT on ImageNet

Figure 5: Metanetwork changes hard to prune network into easy to prune network.

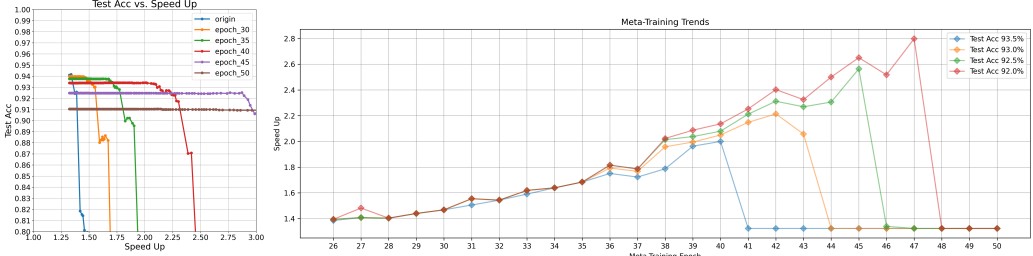

(a) Acc VS. Speed Up curves of metanetworks from different meta-training epochs

(b) Feed network through metanetworks from different meta-training epochs, finetuning, then prune it progressively to find the maximum speed up that can maintain the accuracy above a certain threshold.

Figure 6: Trends in meta-training. (ResNet56 on CIFAR10 as example)

**Finetuning after Metanetwork.** See appendix E.1 for how number of finetuning epochs after metanetwork influence the pruning results.

## 5.4 ABLATION STUDY OF STATISTICS

We compare the statistics between the origin network and the network after metanetwork(finetuning) to find out how metanetwork transforms the network to make it easier to prune. We mainly visualize the $l2$ norm and taylor sensitivity distribution of each layer in Figure 7. Taylor sensitivity here is defined as $w \cdot \Delta L$, it estimates how much the loss will increase if we mask this weight to zero. The larger taylor sensitivity, the more important the weight is and we are unlikely to prune it. More visualization of other statistics are in Appendix H

## 6 EXPERIMENTS ON TRANSFERABILITY AND FLEXIBILITY

### 6.1 FLEXIBLE PRUNING CRITERION

Pruning criterion of our method is highly flexible. We build a series of reasonable pruning criterion (Appendix B.2). At first, we directly use MEAN REDUCE, MAX NORMALIZE, $\alpha = 4$ as default, because this is the same as *group norm* in Fang et al. (2023). Later, we keep everything else the same, and try many different criteria, and find they all works well (Table 2). This demonstrates that our framework is robust and has many flexibilities.

### 6.2 UNSTRUCTURED PRUNING & N:M SPARSITY PRUNING

Most of our experiments are conducted using structured pruning as default. Here we conduct unstructured pruning and N:M sparsity pruning to demonstrate that our methods can be use in all kinds of pruning. We don't use FLOPs to measure the pruning results because unstructured pruning reduces no FLOPs and needs specialized algorithm to accelerate, so we measure the number of parameters instead. See results in Table 3

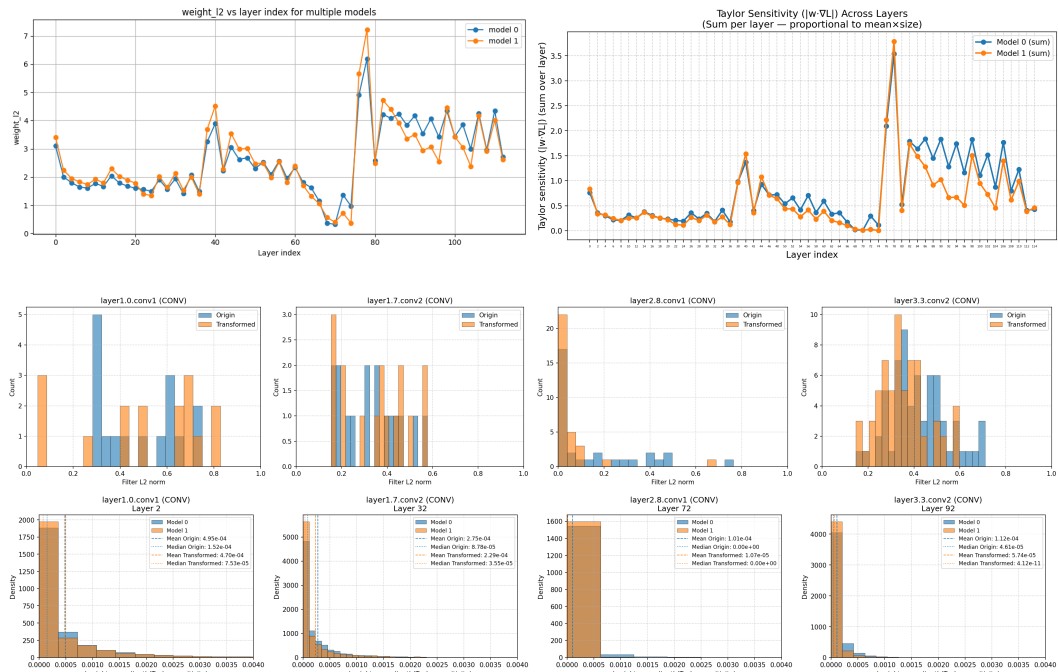

Figure 7: **Statistics:** We compare the statistics of the origin network (orange) and the network after metanetwork and finetuning (blue) when pruning ResNet56 on CIFAR10. The first row is layerwise mean $l_2$ norm and taylor sensitivity. Then we randomly select several layers and visualize their $l_2$ norm distribution in row 2 and taylor sensitivity distribution in row 3. **Observation:** Row 1 shows both norm and sensitivity drops on average, especially in the latter part of network where contains a large amount of redundancy and can be substantially pruned. Row 2 shows $l_2$ norm distribution has been greatly changed. Large norms still exist but more norms tend to be very small. Row 3 shows more parameters become less important under taylor sensitivity.

Table 2: **Flexible pruning criterion:** Pruning ResNet56 on CIFAR10, all experiments use the same origin network wtih Test Acc 93.51%, all criteria use MEAN REDUCE. We tried different values of Alpha and different ways of NORMALIZE, they all work well. Even in the Naive situation, where we use no NORMALIZE and no shrinkage strength (Alpha is 0), the results remain robust.

|  | NORMALIZE | alpha | Pruned Acc | Pruned FLOPs | Speed Up |
|---|---|---|---|---|---|
| Default | MAX | 4 | 93.64% | 65.64% | 2.91 |
| Alpha | MAX | 0 | 93.37% | 65.75% | 2.92 |
|  | MAX | 2 | 92.87% | 68.15% | 3.14 |
|  | MAX | 6 | 93.36% | 65.75% | 2.92 |
| NORMALIZE | MEAN | 4 | 93.42% | 65.52% | 2.90 |
|  | NONE | 4 | 93.08% | 65.64% | 2.91 |
| Naive | NONE | 0 | 93.04% | 65.75% | 2.92 |

Table 3: **Unstructured & N:M sparsity pruning:** Pruning ResNet56 on CIFAR10, meta-train and prune directly with the classical $l_1$ norm. From the results we can see (1) our framework also works great on unstructured pruning. (2) Unstructured pruning outperforms structured pruning because it is more flexible.

| Methods | Pruned Acc | Left Params | Methods | Pruned Acc | Left Params |
|---|---|---|---|---|---|
| Unstructured | 93.95% | 50.25% | Structured(2.9x) | 93.49% | 42,96% |
|  | 94.14% | 20.40% | 3:4 | 94.02% | 75.14% |
|  | 93.43% | 15.42% | 2:4 | 93.97% | 50.27% |
|  | 92.96% | 10.45% | 1:4 | 93.13% | 25.41% |

## 6.3 TRANSFER BETWEEN DATASETS AND ARCHITECTURES

Section 4.4 mentioned that our metanetwork has natural transferability. Following experiments demonstrate that our metanetwork can transfer between similar datasets and network architectures. More relative experiments are in Appendix F.3 F.4

**Transfer between datasets:** 3 datasets, CIFAR10 (Krizhevsky & Hinton, 2009), CIFAR100 (Krizhevsky & Hinton, 2009) and SVHN (Goodfellow et al., 2014) are used. We train metanetwork and the to be pruned network on each of them, traverse through every possible combination, and use the metanetwork to prune the network. Results (Table 4) show that out metanetwork can transfer between similar datasets. See Appendix F.1 for full details **Transfer between architectures:**

Table 4: **Transfer between datasets**: All networks' architecture is ResNet56. Columns represent the training datasets for the metanetwork, and rows represent the training datasets for the to be pruned network. "None" indicates using no metanetwork. Results with metanetwork is obviously better than no metanetwork (The only exception is when training datasets for the to be pruned network is SVHN, and we guess this is because the dataset SVHN itself is too easy).

| Dataset\Metanetwork | CIFAR10 | CIFAR100 | SVHN | None |
|:---:|:---:|:---:|:---:|:---:|
| CIFAR10 | **93.35** | 92.47 | 92.87 | 91.28 |
| CIFAR100 | 69.97 | **70.16** | 69.25 | 68.91 |
| SVHN | 96.79 | 96.50 | **96.86** | 96.78 |

2 architectures, ResNet56 and ResNet110 (He et al., 2016) are used. We train metanetwork and the to be pruned network on each of them, traverse through every possible combination, and use the metanetwork to prune the network. Results (Table 5) show that our metanetwork can transfer between similar datasets. See Appendix F.2 for full details **Possible Future Use:** For large-scale networks,

Table 5: **Transfer between architectures**. All training dataset is CIFAR10. Columns represent the architectures used for training the metanetwork, and rows represent the architecures of the to be pruned network. "None" indicates using no metanetwork. All results with metanetwork is obviously better than no metanetwork.

| Architecture\Metanetwork | ResNet56 | ResNet110 | None |
|:---:|:---:|:---:|:---:|
| ResNet56 | **93.40** | 92.81 | 92.08 |
| ResNet110 | 93.04 | **93.38** | 92.40 |

the metanetwork can be pretrained on smaller architectures of similar design. And when the original training dataset is unavailable or excessively large, a related dataset may be used for pretraining. This transferability substantially enhances the practicality of the framework.

## 7 EXPERIMENTS ON TRANSFORMERS

As mentioned in section 4.4, our framework is theoretically applicable to almost all types of networks with all kinds of pruning. Here we expand our implementations to another widely used arthitecture–transformer (Vaswani et al., 2017). More specifically, we pruned the vision transformer (ViT) from Dosovitskiy et al. (2021) that is trained on ImageNet. See Appendix G for full experiments, including how we convert transformer into graph and conduct further meta-training and pruning.

## 8 CONCLUSION

We propose an entirely new meta-learning framework for network pruning. For a pruning criterion, we use a metanetwork to change a hard to prune network into another easy to prune network for better pruning. This is a general framework that can be theoretically applied to almost all types of networks with all kinds of pruning. We present practical implementations of our framework and achieve outstanding results. Further analysis and experiments show that our framework has natural generality, flexibility, and transferability.

## 9 REPRODUCIBILITY STATEMENT

Our code is available at `https://anonymous.4open.science/r/MetaPruning` together with all the guides to reproduce our experiments.

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

# A COMPARISON WITH PRIOR WORKS

Our framework is entirely new and fundamentally different from all prior works. In this section, we will compare our method with earlier ones to help you better understand the relationship between our method and prior ones, and realize our unique innovations and advantages.

## A.1 GENERAL COMPARISON

Previous pruning approaches can be broadly categorized into fixed pruning methods and learning to prune methods. Our approach falls into the learning to prune category but learns in a completely different way. For clarity, we provide a detailed comparison in Table 6 to help you better understand how our work relates to previous studies.

Table 6: Our framework is entirely new and has many advantages over previous works

|  | Fixed pruning | Learning to prune | Ours |
|---|---|---|---|
| Definition | Pruning with fixed hand-crafted algorithms. | Using neural network learning techniques to learn how to prune. | Also learning to prune, but learning in a completely different way. |
| Performance | Bad. Networks are complex and hand-crafted algorithms are quite limited | Good. | Good (almost best according to our experiments). |
| Cost | Low. No special extra training needed. | High. Need special extra training during each pruning. | Need special extra training before pruning. But once the training is done, can prune as many networks as we want without any special extra training. |
| Potential | Has already been well-studied and commonly used. | The pruning process is too tricky and costly. Can only be used in a very specific situation and lack of generality. | The pruning process is relatively more efficient. Can theoretically be used in almost all situations and has great generality. |

## A.2 COMPUTATIONAL AND MEMORY COSTS

One important question for almost all previous meta-learning approaches is the computational and memory costs are too large. We provide some estimate of memory and computational costs for scaling our methods to larger models and compare with two classic pruning methods–pruning at initalization methods (cheap pruning, hard to training and low accuracy like Lee et al. (2019); Wang et al. (2022)) and iterative magnitude pruning (costly, pruning during training and high performance like Frankle & Carbin (2019); Molchanov et al. (2019)). The results show that our method achieves outstanding results while uses relatively low costs.

For convenience in expression, we refer to pruning at initialization as "init pruning", iterative magnitude pruning as "iter pruning" and our method as "meta pruning".

For all usual networks which satisfy `Edge Number` » `Neuron Number` , the memory and computational cost are all $O(E)$ ($E$ = Edge Number). This is obvious for init and iter pruning. For meta pruning, our metanetwork doesn't scale with the network. So all computation and memory cost is it still $O(E)$.

**PS:** The reason why our metanetwork doesn't scale with the network is that it only learns how to change weights based on local architectures and weights (the range of "local" is fixed for network of different sizes). In all our experiments, it is a small network compared to the to be pruned network,

and it works well. So we have the confidence that it shouldn't and doesn't need to scale with the networks.

While all methods share the same asymptotic cost, their constant factors differ significantly.

Computational cost breaks down as:

$$\text{Compu Cost} = \text{Training cost} + \text{Pruning cost} + \text{Other cost}$$

- **Init pruning:** Low training and pruning costs, negligible other cost.
- **Iter pruning:** High training cost (initial training + multiple finetuning steps) and high pruning cost (multiple pruning rounds), no extra cost.
- **Meta pruning:** Moderate training cost (initial training + 3 finetuning steps), moderate pruning cost (2 pruning steps), plus a small other cost consisting of metanetwork feedforward and amortized meta-training cost per network.

We estimate the relative magnitude in the following table:

| Computation | Training cost | Pruning cost | Other cost |
|:---:|:---:|:---:|:---:|
| Init | 1T | 1P | 0 |
| Iter | > 10T | > 5P | 0 |
| Meta | 2–5T | 2P | $\epsilon$T + AT/N |

Where:
T: Unit training cost
P: Unit pruning cost
$\epsilon$T: Metanetwork feedforward cost (almost zero compared to training cost)
N: Number of networks pruned; meta-training cost is amortized over N (Once we get a metanetwork by meta-training, we can use it to prune as many networks as we want)
A: Meta-training cost. A is estimated 5–20.

Memory cost depends on both the amount of memory used and the duration of usage:

| Method | Amount | Duration | All |
|:---:|:---:|:---:|:---:|
| Init | Small | Short | Low |
| Iter | Medium | Long | Large |
| Meta | Large for a very short time, Small rest of the time | Medium | Medium but requires high memory capacity |

- **Init pruning:** Low memory usage for a short time.
- **Iter pruning:** Medium memory usage for a very long time.
- **Meta pruning:** The "feed forward through metanetwork" process requires large memory usage but takes a very short time. For the rest of the time it uses little memory. It is faster than iter pruning but slower than init pruning. While the average memory cost is moderate, peak usage demands high memory capacity.

Meta-training requires large memory usage for a long time. But its cost can be amortized into each pruning like we mentioned before. Take pruning resnet50 or ViT on ImageNet as example (which is already quite large model and dataset), NVIDIA A100 with 80GiB VRAM is enough for meta-training and feedforward through metanetwork and the rest training and finetuning can be done on NVIDIA RTX 4090 with 24 GiB VRAM. When we don't have enough memory capacity, we can also change the batch size and use more time to make up for the lack of our memory capacity.

In summary, our method achieves outstanding results while uses relatively low costs.

### A.3 GENERALITY

Almost all previous meta-learning based pruning approaches, such as Liu et al. (2019); Li et al. (2020); Wu et al. (2024), are tailored to a specific network and thus lack generality. Because of this they require specialized training for each pruning instance.

Unlike prior methods, our metanetwork learns a universal rule for adapting weights based on local information, thereby enabling more effective pruning without relying on layer-specific or architecture-specific heuristics. We need no special training during each pruning. Once our metanetwork is trained, it can be used to prune as many networks as we want, and even transfer between datasets and architectures.

We divided the generality of learning to prune methods into 4 stages.

(1) Learning once prune one specific network.

(2) Learning once prune one type of networks(same architecture and dataset) as many as we want.

(3) Learning once prune one group of networks(similar architectures and datasets) as many as we want.

(4) Learning once prune any networks.

Here learning refers to learn with extra trainings like gradient descent, reinforcement learning, etc. Finetuning or other fixed rules processes are not included. As far as we know, all prior learning to prune methods only reach stage (1). Our method reaches stage (3) and shows great improvement in generality over prior works. Follow our ideas and pretrain the metanetwork on various networks and datasets may provide a possible way to state (4), whether this will work requires further exploration in the future.

In summary, our method shows great improvement in generality over prior learning to prune methods.

### A.4 A CONCRETE EXAMPLE

We provide a concrete example compared with prior works and report everything-time, hardware, gpu memory, results, etc. to give readers a more intuitive understanding of our method.

We compare our work with Fang et al. (2023) on pruning ResNet56 on CIFAR10. We choose Fang et al. (2023) because we want to compare our work with learning to prune methods in recent years that also have strong results like us. The best candidates are Fang et al. (2023) and Wu et al. (2024). While Wu et al. (2024) is a pruning before training method, both Fang et al. (2023) and our work are pruning after training methods, so we choose Fang et al. (2023). The key idea of Fang et al. (2023) is sparsity training before pruning. It does special training on the network before pruning to make it more sparse and easier to prune.

All experiments are run on 1 NVIDIA RTX 4090. See table 7 for time consumptions of dfferent methods. To align with Fang et al. (2023), we don't use init pruning, and target at a speed up of 2.5x, which is the largest speed up in paper Fang et al. (2023). In ours(full), we finetune 100 epochs after metanetwork and 100 epochs after pruning. But later we found our method is stronger and the speed up 2.5x isn't that hard. So in ours(efficient), we finetune 60 epochs after metanetwork and 60 epochs after pruning but also get results comparable to Fang et al. (2023). The DepGraph experiments use their default settings, sparsity training 100 epochs before pruning and finetune 100 epochs after pruning.

Table 7: Time (miniutes)

| Process | Ours(full) | Ours(efficient) | DepGraph |
|---|---|---|---|
| Meta-Train | 192 + 165 = 357 | 192 + 165 = 357 | 0 |
| Prune & Finetune | 67 | 43 | 84 |

From the results we can see our methods is quite efficient. It uses 357 minutes for meta-train, which is in a resonable range. We can greatly reduce time consumption in this process with some experiences, but for the sake of fairness in comparison, we must pretend we have no relevant experience. Among the 357 minutes, 192 minutes are used for generating 100 epochs metanetworks and 165 minutes are used for visualizing the Acc VS. Speed Up Curve to select the appropriate metanetwork for pruning. Once a metanetwork is trained, it can be used to prune as many networks as we want,

so we can amortize the time for meta-training into each pruning. Figure 8 shows the amortized time consumption of our work compared to DepGraph. In all, our work is more powerful and more efficient in time if we prune several more networks or already have a trained metanetwork.

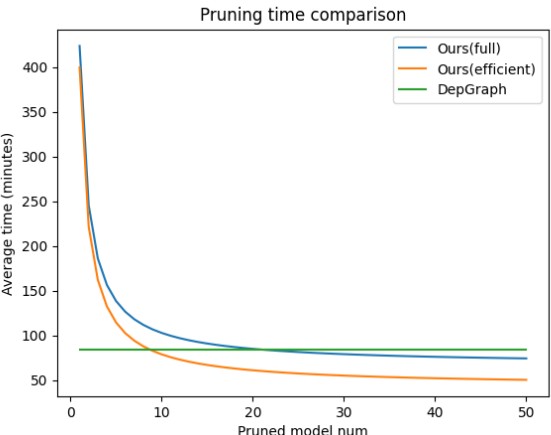

Figure 8: Amortized Time Consumption

As we mentioned in section A.2, our methods use a large VRAM for a long time in meta-training, a still large but relatively smaller VRAM for a very short time when feed forward through metanetwork during pruning, and a small VRAM for the standard finetuning. Specifically in this experiment, we use 10000 MiB for meta-training, 6000 MiB for feedforward through metanetwork, and 1000 MiB for finetuning. NVIDIA A100 with 80 GiB VRAM is enough for meta-training large models like ResNet50 and ViT-B-16. When we don't have enough memory capacity, we can also change the batch size and use more time to make up for the lack of our memory capacity. We've also tried reducing the size of metanetwork and fit on NVIDIA A100 40G and the results seem not much influenced. All finetuning are standard finetuning and can be done on NVIDIA RTX 4090 with 24G VRAM. In all, during pruning only the feed forward through metanetwork requires large VRAM for a very short time and anything else is plain finetuning and requires small VRAM, during meta-training the VRAM is in a resonable range.

## B  Framework implementation details

### B.1  Metanetwork(GNN) Architecture

We build our metanetwork based on the message passing framework (Gilmer et al., 2017) and PNA (Corso et al., 2020) architecture.

Notation:

$$a \leftarrow b \quad : \text{assignment/update, overwrite } a \text{ with the value of } b \tag{10}$$

$$X \odot Y \quad : \text{Hadamard/elementwise product, } (X \odot Y)i = X_i Y_i \tag{11}$$

We use $v_i$ to represent node $i$'s feature vector, and $e_{ij}$ to represent feature vector of the edge between $i$ and $j$. We consider undirected edges, i.e., $e_{ij}$ is the same as $e_{ji}$. Our metanetwork is as follows.

First, all input node and edge features are encoded into the same hidden dimension.

$$v \leftarrow \text{MLP}_{NodeEnc}(v^{in}), \tag{12}$$

$$e \leftarrow \text{MLP}_{EdgeEnc}(e^{in}). \tag{13}$$

Then they will pass through several message passing layers. In each layer, we generate the messages:

$$m_{ij} \leftarrow \text{MLP}^1_{Node}(v_i) \odot \text{MLP}^2_{Node}(v_j) \odot e_{ij}, \tag{14}$$

$$m'_{ij} \leftarrow \text{MLP}^1_{Node}(v_j) \odot \text{MLP}^2_{Node}(v_i) \odot (e_{ij} \odot EdgeInvertor), \tag{15}$$

where $m_{ij}$ and $m'_{ij}$ respectively encode message from $i$ to $j$ and message from $j$ to $i$, and EdgeInvertor is defined as:

$$EdgeInvertor \triangleq [\underbrace{1, 1, \ldots, 1}_{\text{hidden\_dim}/2}, \underbrace{-1, -1, \ldots, -1}_{\text{hidden\_dim}/2}]. \tag{16}$$

The intuition behind EdgeInvertor is to let the first half dimensions to learn undirectional features invariant to exchanging $i, j$, and the second half to capture directional information that changes equivariantly with reverting $i, j$. Empirically we found this design the most effective among various ways to encode edge features.

Then, we aggregate the messages to update node features:

$$v_i \leftarrow v_i + \text{PNA}_{Aggr}(m_{ij}) + \text{PNA}_{Aggr}(m'_{ij}), \tag{17}$$

where we have

$$\text{PNA}_{Aggr}(m_{ij}) \triangleq \text{MLP}_{Aggr}\left(\left[\underset{j:(i,j)\in E}{\text{MEAN}}(m_{ij}), \underset{j:(i,j)\in E}{\text{STD}}(m_{ij}), \underset{j:(i,j)\in E}{\text{MAX}}(m_{ij}), \underset{j:(i,j)\in E}{\text{MIN}}(m_{ij})\right]\right). \tag{18}$$

For edge features in each layer, we also update them by:

$$e_{ij} \leftarrow e_{ij} + \text{MLP}^1_{Edge}(v_i) \odot \text{MLP}^2_{Edge}(v_j) \odot e_{ij}$$
$$+ \text{MLP}^1_{Edge}(v_j) \odot \text{MLP}^2_{Edge}(v_i) \odot (e_{ij} \odot EdgeInvertor). \tag{19}$$

Finally, we use a decoder to recover node and edge features to their original dimensions:

$$v_{pred} \leftarrow \text{MLP}_{NodeDec}(v), \tag{20}$$

$$e_{pred} \leftarrow \text{MLP}_{EdgeDec}(e). \tag{21}$$

We multiply predictions by a residual coefficient and add them to the original inputs as our final outputs:

$$v_{out} = \alpha \cdot v_{pred} + v_{in}, \tag{22}$$

$$e_{out} = \beta \cdot e_{pred} + e_{in}, \tag{23}$$

where $\alpha$ and $\beta$ in practice are set to a small real numbers like 0.01. This design enables the metanetwork to learn only the delta weights on top of the original weights, instead of a complete reweighting.

## B.2 PRUNING CRITERION

### B.2.1 DEPGRAPH AND TORCH-PRUNING

DepGraph (Fang et al., 2023) proposes a general sparsity regularization based structural pruning framework that can be applied to a wide range of neural network architectures, including CNNs, RNNs, GNNs, Transformers, etc. Alongside the paper, the authors released **Torch-Pruning**, a powerful Python library that enables efficient structural pruning for most modern architectures. In our work, the pruning criterion is designed based on the methodology of DepGraph (Fang et al., 2023), and all pruning operations are implemented using Torch-Pruning.

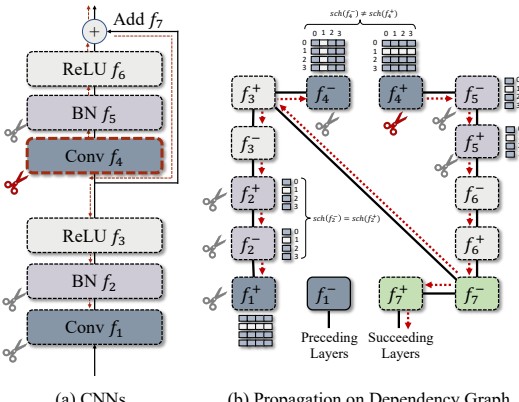

(a) CNNs  (b) Propagation on Dependency Graph

Figure 9: A picture from (Fang et al., 2023). It shows how we group parameters together in structural pruning.

### B.2.2 STRUCTURAL PRUNING

We perform most of our experiments by default using structural pruning, but also include a small number of experiments with unstructured and N:M sparsity pruning to demonstrate that our method can be applied to all kinds of pruning. Here we mainly introduce the structural pruning.

Structural pruning involves removing parameters in predefined *groups*, so that the resulting pruned model can be used as a standalone network—without relying on the original model with masks, nor requiring specialized AI accelerators or software to realize reductions in memory footprint and computational cost.

In structural pruning, a *pruning group* consists of all parameters that must be pruned together to maintain network consistency. For example, in Figure 9(a), if an input channel of convolution layer $f_4$ is pruned, the corresponding channel in batch normalization (BN) layer $f_2$ and the output channel of convolution layer $f_1$ must be pruned as well. Furthermore, due to the presence of a residual connection, the corresponding channels in BN layer $f_5$ and the output channel of convolution layer $f_4$ must also be removed. All of these channels together form a single pruning group. We define the *number of prunable dimensions* of a group as the size of the input channel dimension of Conv $f_4$ (which is equivalent to the corresponding dimensions in BN $f_2$, the output channel of Conv $f_1$, etc.)

For a more comprehensive theory to find all pruning groups in structural pruning, refer to the DepGraph (Fang et al., 2023) paper (section 3.1 & 3.2).

### B.2.3 A IMPORTANCE SCORE FUNCTION

Given a parameter group $g = \{w_1, w_2, \ldots, w_{|g|}\}$ with K prunable dimensions indexed by $w_t[k]$ ($t \in \{1, 2 \ldots |g|\}$), the score of the k th prunable dimension in group g is written is $I_{g,k}$, we introduce a general way to generate a series types of importance scores. The way has two key concepts, REDUCE and NORMALIZE. We first use REDUCE to reduce scores of parameters in the same group and in the same prunable dimension into one score.

$$I_{t,k} = \operatorname*{REDUCE}_{t:t\in\{1,2\ldots|g|\}} \left(|w_t[k]|^p\right) \tag{24}$$

Here p is a hyperparameter and we usually set it to 2, REDUCE can be (but is not limited to) the following:

(1) **MEAN:** $(w_1 + w_2 + ... + w_{|g|})/|g|$

(2) **FIRST:** $w_1$

Then we use NORMALIZE to normalize scores in the different prunable dimensions of the same pruning group.

$$\hat{I}_{g,k} = I_{g,k}/\underset{k:k\in\{1,2...K\}}{\text{NORMALIZE}}(I_{g,k}) \qquad (25)$$

Here NORMALIZE can be (but is not limited to) the following:

(1) **NONE:** 1 (use no normalize).

(2) **MEAN:** $(I_{g,1} + I_{g,2} + ... + I_{g,K})/K$

(3) **MAX:** The maximum amoung $I_{g,1}, I_{g,2}...I_{g,K}$

We calculate scores of all prunable dimensions in all groups in a network, rank them and prune according to their scores from lower to higher.

### B.2.4 A SPARSITY LOSS

During training, our sparsity loss is defined as :

$$\mathcal{R}(g,k) = \sum_{k=1}^{K} \gamma_k \cdot \hat{I}_{g,k} \qquad (26)$$

Where $\gamma_k$ refers to a shrinkage strength applied to the parameters to modify the gradients for better training. Defined as :

$$\gamma_k = 2^{\alpha \frac{\sqrt{I_g^{\max}} - \sqrt{I_{g,k}}}{\sqrt{I_g^{\max}} - \sqrt{I_g^{\min}}}} \qquad (27)$$

Here $\alpha$ can be but is not limited to:

(1) **4:** What we use as default in all our experiments.

(2) **0:** Same as using no shrinkage strength.

In sparsity loss, we treat $\gamma_k$ and NORMALIZE in the denominator of $\hat{I}_{g,k}$ simply as constants, which means gradients aren't backpropagated through them. Gradients only back propagated from $\mathcal{R}(g,k)$ to $I_{g,k}$ to the parameters of the network.

In our code, rather than calculate the sparsity loss, we directly modify the gradients of the parameters of the network, which get the same results.

### B.2.5 KINDS OF PRUNING CRITERION

By choosing different ways of REDUCE and NORMALIZE (appendix B.2.3) and different $\alpha$ (appendix B.2.4), we can make different pruning cirteria in a uniform framework. **In most of our experiments, we use MEAN REDUCE, MAX NORMALIZE, $\alpha = 4$, as our default pruning criterion, and we name it *group $\ell_2$ norm max normalizer*.** We also tried different criteria in our experiments and found they are almost all effective, which further shows the flexibility and effectiveness of our framework.

### B.2.6 PAY ATTENTION

Though we draw ideas from Fang et al. (2023) and use their Torch-Pruning libirary, we find their paper is inconsistent with their code. We wrote appendix B.2.3 & B.2.4 strictly based on their code, which is the Torch-Pruning library. If you read the Fang et al. (2023) paper section 3.3, you may find our description a little different from theirs, because their paper is inconsistent with their code, and we are in line with their code.

## C    EXPERIMENTAL PRELIMINARIES

### C.1    TERMINOLOGIES

**Speed Up:** In line with prior works, we define **Speed Up ≜ Origin FLOPs / Pruned FLOPs**. It reflects the extent to which our pruning reduces computation, thereby accelerating the network's operation.

**Acc vs. Speed Up Curve:** We generate the curve by prune the network little by little. After each pruning step, we evaluated the model's performance on the test set (no finetuning in this step) and recorded a data point representing the current speed up and corresponding test accuracy. Connecting these points forms a curve that illustrates how accuracy changes as the speed up increases.

### C.2    PRUNING PIPELINE

Our pruning pipeline can be summarized as:

$$\underbrace{\text{Initial Pruning (Finetuning)}}_{\text{Optional}} \rightarrow \underbrace{\text{Metanetwork (Finetuning)} \rightarrow \text{Pruning (Finetuning)}}_{\text{Necessary}} \quad (28)$$

**Initial Pruning:** Empirically, we observe that the metanetwork performs better on hard to prune origin networks. However, many networks are initially easy to prune. For instance, ResNet56 on CIFAR10 can be pruned with a $1.3\times$ speedup and then finetuned without any accuracy loss. Therefore, unless stated otherwise, we apply the initial pruning step in all our experiments—both when generating meta-training data models and during pruning. In this step, we slightly prune the original network with a fixed speed up (a hyperparameter that can be determined in a few quick trials) to make it harder to prune without loss in accuracy (Figure 10a), thereby exploiting the full potential of the metanetwork. This effect is analogous to removing low-quality samples from a dataset in order to obtain a higher-quality subset: an easy to prune network contains many low-quality parameters, which may harm the training of the metanetwork. This step isn't necesary, but facilitates convergence, saves VRAM use, and helps achieve better pruning results (Figure 10b) during most of the time.

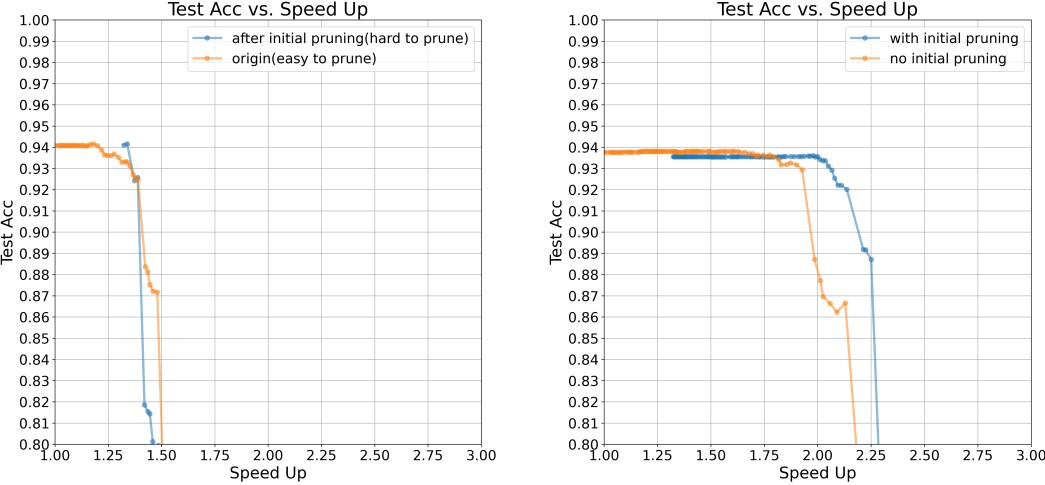

(a) The origin network is easy to prune, after initial pruning and slight finetuning, it becomes hard to prune.

(b) Initial pruning helps get better metanetworks and better pruning results

Figure 10: With or without initial pruning (ResNet56 on CIFAR10 as an example)

**Metanetwork:** Change the origin hard to prune network into a graph, feed it through the metanetwork to get a new graph, and change it back into a new network.

**Pruning:** Prune the new easy to prune network.

**Finetuning:** Train the network with a relatively small learning rate. This is a common step that is widely used in pruning tasks.

Unlike prior learning to prune methods that require special training during each pruning, our pruning pipeline only requires a feed forward through metanetwork and standard finetuning, which is simple, general and effective.

Once we have a metanetwork, we can prune as many networks as we want. All our pruning models are completely different from meta-training data models. This demonstrates our metanetwork naturally has great transferability and no prior work has done something like this before.

# D    EXPERIMENTAL ON CNNs

## D.1    GENERAL EXPERIMENT SETUP

### D.1.1    GENERAL SETTINGS

**Optimizer:** For all data model training and fine-tuning, we use `torch.optim.SGD` with `momentum=0.9`. For meta-training, we employ `torch.optim.AdamW`. All learning rates are controlled using `torch.optim.lr_scheduler.MultiStepLR` with `gamma=0.1`. The learning rate is adjusted at specified milestones; for instance, if `milestones=[50, 90]`, the learning rate is multiplied by 0.1 at epochs 50 and 90. These settings are not strictly necessary—other optimizers and schedulers may be equally effective.

**Data for Meta-Training:** As discussed in Section 4.2, meta-training requires two types of data: data models and traditional training datasets. To generate a data model, we train a base model, perform initial pruning followed by finetuning, and finally save the model along with relevant information. Typically, generating 2–10 data models is sufficient, with 1–8 used for meta-training and the remainder reserved for meta-evaluation or testing.

**Meta Evaluation (Meta Eval):** Meta evaluation is a process used to assess the quality of the metanetwork during meta-training. **We did not include it in our final experiments due to its high computational cost; however, we retain it as an optional tool for future research.** At the end of each meta-training epoch, we evaluate the metanetwork by feeding unseen data models (not used in meta-training) through the network, finetuning, and then pruning to achieve the maximum speed-up while maintaining accuracy above a predefined threshold. A higher resulting speed-up indicates better metanetwork performance.

**Visualizing Metanetworks: This is a key technique used throughout our experiments.** When we refer to visualizing a metanetwork, we mean passing a model through the metanetwork, followed by finetuning, and then visualize the "Acc VS. Speed-Up" curve of the resulting model, as illustrated in Figure 5. This visualization provides insights into the metanetwork's behavior and helps determine whether it is suitable for pruning. An ideal metanetwork should exhibit a long flat region in the curve where accuracy remains close to or above the target pruning accuracy. There are two ways to select a suitable metanetwork: (1) using meta-evaluation during meta-training, or (2) visualizing metanetworks post-training. In our final experiments, we exclusively use the second method, which is significantly more efficient. We typically do not visualize all metanetworks but instead search for the best one using a binary search strategy.

**Relationship between "Acc VS. Speed Up Curve" and Pruning Performance:** Empirically, pruning performance can be qualitatively predicted from the "Accuracy VS. Speed Up" curve. The accuracy in the flat region of the curve typically represents the maximum achievable accuracy; after pruning and finetuning, the final accuracy is usually the same or slightly lower, and sometimes only a little bit higher at most. The amount of speed up that preserves this top accuracy is generally larger than the speed up at the curve's turning point, and these two values tend to correlate—the larger the turning point speed up, the larger the maintainable speed up. Therefore, if we aim for a pruned model with, for example, 93% accuracy, we should select a metanetwork whose flat-region accuracy is around or slightly above 93%, and whose flat region is as long as possible. We cannot give a teoretical guarantee between the "Acc VS. Speed Up" curve and the pruning performance. But this is not only our problem, because as far as we know, all other pruning methods can't give theoretical gurarantee between their pruning criterion and final performance as well. The reasons can be the network itself is too complex and processes like finetuning changes everything in an unpredictable way.

**Big Batch Size and Small Batch Size:** We use two different batch sizes in our experiments: a small batch size and a big batch size. The big batch size is used during meta-training to compute accuracy loss on new networks. Due to memory constraints, such large batches cannot be processed in a single forward and backward pass. Instead, we accumulate gradients over multiple smaller mini-batches before performing a single optimizer update—a common PyTorch practice. The big batch size is thus expressed as `batchsize * iters`, indicating that one `optimizer.step()` is performed after `iters` forward/backward passes, each using a mini-batch of size `batchsize`. For all other standard training and finetuning tasks, we use the small batch size.

### D.1.2 GENERAL META TRAINING DETAILS

**Equivalent Conversion between Network and Graph:** During meta-training, we first convert the original network into an origin graph. Then, we feed this origin graph through the metanetwork to generate a new graph. Finally, we convert the new graph back into a new network. When converting the original network into the origin graph, all network parameters are mapped to either node or edge features in the graph. However, some graph features do not correspond to any network parameter. For these missing features, we use default values that effectively simulate the absence of those parameters. When converting the new graph back into the new network, we only map those trainable parameters from the new graph back to the new network. For untrainable parameters in the new network, we keep it the same as the old network. We provide more explanations for this conversion process in the following subsections: D.2, D.3, and D.4.

**The Relative Importance of Sparsity Loss and Accuracy Loss:** In our experiments, we introduce a hyperparameter called **"pruner reg"**, which controls the relative importance of sparsity loss compared to accuracy loss. During backpropagation, the gradient from the sparsity loss is scaled(multiplied) by this "pruner reg" value, while the gradient from the accuracy loss remains unscaled.

**Meta Training Milestone:** As discussed in the Ablation Study (Section 5.3), as the number of training epochs increases, the metanetwork's ability to make the network easier to prune becomes stronger, while its ability to maintain the accuracy becomes weaker. During meta-training, we usually set a milestone for our learning rate scheduler. Before this milestone, we use a relatively large learning rate to quickly improve the metanetwork's ability to make networks easier to prune. After the milestone, we switch to a smaller learning rate to finetune the metanetwork and slow down the change of the two abilities, allowing us to select a well-balanced metanetwork. To determine a reasonable milestone value, we can first perform a short meta-training phase using the large learning rate, then visualize the resulting metanetworks to identify the point where the accuracy-maintaining ability of metanetwork is a bit stronger than we expect.

**Choose the Appropriate Metanetwork:** During meta-training, we save the metanetwork after each epoch. After training is complete, we search for the most suitable metanetwork by visualizing its performance using a binary search strategy. Specifically, we start by visualizing the metanetwork from the middle epoch. If the accuracy of its flat region is below our target pruned accuracy, we next visualize the metanetwork from the first quarter of training. Otherwise, we check the one from the third quarter. We continue this process iteratively until we find a metanetwork that meets our criteria: a relatively high accuracy in the flat region and a long flat region indicating robustness to pruning.

### D.1.3 GENERAL PRUNING DETAILS :

**Pruning Speed Up:** Once a suitable metanetwork has been selected for pruning, the next step is to determine the target speed up. In general, after visualizing the metanetwork, we observe a flat region followed by a sharp decline in accuracy. We define a *turning point* as the speed up at which the accuracy drops below a certain threshold. Experimentally, we find that we can safely prune the model at a speed up slightly higher than this turning point without incurring any accuracy loss after finetuning. For instance, when pruning ResNet-56 on CIFAR-10, we consider an accuracy drop below 0.93 as the turning point (typically around $2.0\times$ speed up), and in practice, we are able to achieve a $2.9\times$ speed up with almost no loss in accuracy after finetuning.

## D.2  RESNET56 ON CIFAR10

### D.2.1  EQUIVALENT CONVERSION BETWEEN NETWORK AND GRAPH

We change ResNet56 into a graph with node featrues of 8 dimensions and edge features of 9 dimensions.

**The node features** consist of 4 features derived from the batch normalization parameters (weight, bias, running mean, and running variance) of the previous layer, along with 4 features from the batch normalization of the previous residual connection. If no such residual connection exists, we assign default values $[1, 0, 0, 1]$ to the corresponding 4 features.

**The edge features** are constructed by zero-padding all convolutional kernels to a size of $3 \times 3$ (note that our networks only contain kernels of size $3 \times 3$ or $1 \times 1$), and then flattening them into feature vectors. We treat all residual connections as convolutional layers with kernel size $1 \times 1$. For example, consider two layers A and B connected via a residual connection, with neurons indexed as $1, 2, 3, \ldots$ in both layers. If the residual connection has no learnable parameters (i.e., it directly adds the input to the output), we represent the edge from $A_i$ to $B_i$ as a $1 \times 1$ convolutional kernel with value 1, and the edge from $A_i$ to $B_j$ (where $i \neq j$) as a $1 \times 1$ kernel with value 0. If the residual connection includes parameters (e.g., a downsample with a convolutional layer and batch normalization), we construct the edge features in the same way as for standard convolutional layers.

### D.2.2  META TRAINING

**Data Models :** We generate 10 models as our data models, among them, 8 are used for meta-training and 2 are used for validation (visualize the metanetwork). When generating each data model, we train them 200 epochs with learning rate 0.1 and milestone "100, 150, 180", then pruning with speed up 1.32x, followed by a finetuning for 80 epochs with learning rate 0.01 and milestone "40, 70". Finally, we can get a network with accuracy around 93.5%.

**One Meta Training Epoch :** Every epoch we enumerate over all 8 data models. When enumerate a data model, we feedforward it through the metanetwork and generate a new network. We feed all our CIFAR10 training data into the new network to calculate the accuracy loss, and use the parameters of new network to calculate the sparsity loss. Then we backward the gradients from both two losses to update our metanetwork.

**Training:**  We train our metanetwork with learning rate 0.001, milestone "3", weight decay 0.0005 and pruner reg 10. Finally we use metanetwork from epoch 39 as our final network for pruning.

### D.2.3  GPU USAGE

All tasks can be run on 1 NVIDIA RTX 4090.

### D.2.4  FULL RESULTS

All results are summarized in Table 8, where we repeat the pruning process 3 times using different seeds: 7, 8, 9. It is important to note that due to the design of our algorithm and implementation, we cannot prune the network to exactly the same speed up across different runs. For instance, when targeting a $1.3\times$ speed up, the actual achieved speed up may be slightly larger, such as $1.31\times$, $1.32\times$, or $1.35\times$.

The aggregated statistical results are presented in Table 9, where each value is reported in the form of `mean(standard deviation)`.

### D.2.5  HYPERPARAMETERS

See Table 10.

Table 8: ResNet56 on CIFAR10 Full

| Method | Base | Pruned | Δ Acc | Pruned FLOPs | Speed Up |
|---|---|---|---|---|---|
| NISP (Yu et al., 2018) | — | — | — | 43.2% | 1.76 |
| Geometric (He et al., 2019b) | 93.59 | 93.26 | -0.33 | 41.2% | 1.70 |
| Polar (Zhuang et al., 2020) | 93.80 | 93.83 | 0.03 | 46.8% | 1.88 |
| DCP-Adapt (Zhuang et al., 2018) | 93.80 | 93.81 | 0.01 | 47.0% | 1.89 |
| CP (Li et al., 2017a) | 92.80 | 91.80 | -1.00 | 50.0% | 2.00 |
| AMC (He et al., 2018b) | 92.80 | 91.90 | -0.90 | 50.0% | 2.00 |
| HRank (Lin et al., 2020a) | 93.26 | 92.17 | -1.09 | 50.0% | 2.00 |
| SFP (He et al., 2018a) | 93.59 | 93.36 | -0.23 | 52.6% | 2.11 |
| ResRep (Ding et al., 2021) | 93.71 | 93.71 | 0.00 | 52.8% | 2.12 |
| SCP (Kang & Han, 2020) | 93.69 | 93.23 | -0.46 | 51.5% | 2.06 |
| FPGM (He et al., 2019a) | 93.59 | 92.93 | -0.66 | 52.6% | 2.11 |
| FPC (He et al., 2020) | 93.59 | 93.24 | -0.35 | 52.9% | 2.12 |
| DMC (Gao et al., 2020) | 93.62 | 92.69 | -0.93 | 50.0% | 2.00 |
| GNN-RL (Yu et al., 2022) | 93.49 | 93.59 | 0.10 | 54.0% | 2.17 |
| DepGraph w/o SL (Fang et al., 2023) | 93.53 | 93.46 | -0.07 | 52.6% | 2.11 |
| DepGraph with SL (Fang et al., 2023) | 93.53 | **93.77** | 0.24 | 52.6% | 2.11 |
| ATO (Wu et al., 2024) | 93.50 | 93.74 | 0.24 | 55.0% | 2.22 |
| Meta-Pruning (ours) | 93.51 | 93.64 | 0.13 | **56.5%** | **2.30** |
| Meta-Pruning (ours) | 93.51 | 93.75 | 0.24 | 58.0% | 2.38 |
| Meta-Pruning (ours) | 93.51 | **93.78** | 0.27 | **56.5%** | **2.30** |
| GBN (You et al., 2019a) | 93.10 | 92.77 | -0.33 | 60.2% | 2.51 |
| AFP (Ding et al., 2018) | 93.93 | 92.94 | -0.99 | 60.9% | 2.56 |
| C-SGD (Ding et al., 2019) | 93.39 | 93.44 | 0.05 | 60.8% | 2.55 |
| Greg-1 (Wang et al., 2021) | 93.36 | 93.18 | -0.18 | 60.8% | 2.55 |
| Greg-2 (Wang et al., 2021) | 93.36 | 93.36 | 0.00 | 60.8% | 2.55 |
| DepGraph w/o SL (Fang et al., 2023) | 93.53 | 93.36 | -0.17 | 60.2% | 2.51 |
| DepGraph with SL (Fang et al., 2023) | 93.53 | **93.64** | 0.11 | 61.1% | 2.57 |
| ATO (Wu et al., 2024) | 93.50 | 93.48 | -0.02 | 65.3% | 2.88 |
| Meta-pruning (ours) | 93.51 | **93.64** | 0.13 | 65.6% | 2.91 |
| Meta-pruning (ours) | 93.51 | 93.28 | -0.23 | **65.9%** | **2.93** |
| Meta-pruning (ours) | 93.51 | **93.49** | -0.02 | **66.0%** | **2.94** |
| Meta-pruning (ours) | 93.51 | **93.27** | -0.24 | 66.8% | 3.01 |
| Meta-pruning (ours) | 93.51 | 93.10 | -0.41 | **66.9%** | **3.02** |
| Meta-pruning (ours) | 93.51 | **93.52** | 0.01 | **67.0%** | **3.03** |

Table 9: ResNet56 on CIFAR10 Statistics

| Pruned Accuracy | Pruned FLOPs | Speed Up |
|---|---|---|
| 93.72(±0.06) | 57.0%(±0.71%) | 2.3267(±0.0377) |
| 93.47(±0.15) | 65.83%(±0.17%) | 2.9267(±0.0125) |
| 93.30(±0.17) | 66.90%(±0.08%) | 3.0200(±0.0081) |

Table 10: ResNet56 on CIFAR10 Hyperparameters

| Types | Name | Value |
|---|---|---|
| **Compute Resources** | GPU | NVIDIA RTX 4090 |
| | parallel | No |
| **Batch size** | small batch size | 128 |
| | big batch size | $500 \times 100$ |
| **Prepare Data Models** | data model num | 10 ( 8 + 2 ) |
| Train From Scratch | epoch | 200 |
| | lr | 0.1 |
| | weight decay | 0.0005 |
| | milestone | "100, 150, 180" |
| Initial Pruning | speed up | 1.32 |
| Finetuning | epoch | 80 |
| | lr | 0.01 |
| | weight decay | 0.0005 |
| | milestone | "40, 70" |
| **Metanetwork** | num layer | 8 |
| | hiddim | 64 |
| | in node dim | 8 |
| | in edge dim | 9 |
| | node res ratio | 0.01 |
| | edge res ratio | 0.01 |
| **Meta Training** | lr | 0.001 |
| | weight decay | 0.0005 |
| | milestone | "3" |
| | pruner reg | 10 |
| **Final Pruning** | metanetwork epoch | 39 |
| | speed up | 2.3, 2.9, 3.0 |
| Finetuning After Metanetwork | epoch | 100 |
| | lr | 0.01 |
| | weight decay | 0.0005 |
| | milestone | "60, 90" |
| Finetuning After Pruning | epoch | 140 |
| | lr | 0.01 |
| | weight decay | 0.0005 |
| | milestone | "80, 120" |

### D.3    VGG19 on CIFAR100

#### D.3.1    Equivalent Conversion between Network and Graph

We change VGG19 into a graph with node featrues of 5 dimensions and edge features of 9 dimensions.

**The node features** consist of 4 features derived from the batch normalization parameters (weight, bias, running mean, and running variance) of the previous layer and 1 feature derived from the bias of previous layer (0 if bias doesn't exist).

**The edge features** are constructed by zero-padding all convolutional kernels to a size of $3 \times 3$ (note that our networks only contain kernels of size $3 \times 3$ or $1 \times 1$), and then flattening them into feature vectors.

#### D.3.2    Meta Training

**Data Models :** We generate 10 models as our data models, among them, 8 are used for meta-training and 2 are used for validation (visualize the metanetwork). When generating each data model, we train them 200 epochs with learning rate 0.1 and milestone "100, 150, 180", then pruning with speed up 2.0x, followed by a finetuning for 140 epochs with learning rate 0.01 and milestone "80, 120". Finally, we can get a network with accuracy around 73.5%.

**One Meta Training Epoch :** Every epoch we enumerate over all 8 data models. When enumerate a data model, we feedforward it through the metanetwork and generate a new network. We feed all our CIFAR100 training data into the new network to calculate the accuracy loss, and use the parameters of new network to calculate the sparsity loss. Then we backward the gradients from both two losses to update our metanetwork.

**Training:**  We train our metanetwork with learning rate 0.001, milestone "10", weight decay 0.0005 and pruner reg 10. Finally we use metanetwork from epoch 38 as our final network for pruning.

#### D.3.3    GPU Usage

All tasks can be run on 1 NVIDIA RTX 4090.

#### D.3.4    Full Results

All results are summarized in Table 11, where we repeat the pruning process 3 times using different seeds: 7, 8, 9. It is important to note that due to the design of our algorithm and implementation, we cannot prune the network to exactly the same speed up across different runs. For instance, when targeting a $8.90\times$ speed up, the actual achieved speed up may be slightly larger, such as $8.95\times$, $9.01\times$, or $9.02\times$.

The aggregated statistical results are presented in Table 12, where each value is reported in the form of `mean(standard deviation)`.

#### D.3.5    Hyperparameters

See Table 13.

Table 11: VGG19 on CIFAR100 Full

| Method | Base | Pruned | Δ Acc | Pruned FLOPs | Speed Up |
|---|---|---|---|---|---|
| OBD (Wang et al., 2019) | 73.34 | 60.70 | -12.64 | 82.55% | 5.73 |
| OBD (Wang et al., 2019) | 73.34 | 60.66 | -12.68 | 83.58% | 6.09 |
| EigenD (Wang et al., 2019) | 73.34 | 65.18 | -8.16 | 88.64% | 8.80 |
| Greg-1 (Wang et al., 2021) | 74.02 | 67.55 | -6.67 | 88.69% | 8.84 |
| Greg-2 (Wang et al., 2021) | 74.02 | 67.75 | -6.27 | 88.69% | 8.84 |
| DepGraph w/o SL (Fang et al., 2023) | 73.50 | 67.60 | -5.44 | 88.73% | 8.87 |
| DepGraph with SL (Fang et al., 2023) | 73.50 | **70.39** | -3.11 | 88.79% | 8.92 |
| Meta-Pruning (ours) | 73.65 | 68.65 | -5.00 | 88.81% | 8.94 |
| Meta-Pruning (ours) | 73.65 | 67.63 | -6.02 | **88.96%** | **9.06** |
| Meta-Pruning (ours) | 73.65 | **69.75** | -3.90 | **88.83%** | **8.95** |

Table 12: VGG19 on CIFAR100 Statistics

| Pruned Accuracy | Pruned FLOPs | Speed Up |
|---|---|---|
| 68.68(±0.87) | 88.87%(±0.07%) | 8.9833(±0.0544) |

Table 13: VGG19 on CIFAR100 Hyperparameters

| Types | Name | Value |
|---|---|---|
| **Compute Resources** | GPU | NVIDIA RTX 4090 |
| | parallel | No |
| **Batch size** | small batch size | 128 |
| | big batch size | $500 \times 100$ |
| **Prepare Data Models** | data model num | 10 ( 8 + 2 ) |
| Train From Scratch | epoch | 200 |
| | lr | 0.1 |
| | weight decay | 0.0005 |
| | milestone | "100, 150, 180" |
| Initial Pruning | speed up | 2.0 |
| Finetuning | epoch | 140 |
| | lr | 0.01 |
| | weight decay | 0.0005 |
| | milestone | "80, 120" |
| **Metanetwork** | num layer | 8 |
| | hiddim | 32 |
| | in node dim | 5 |
| | in edge dim | 9 |
| | node res ratio | 0.05 |
| | edge res ratio | 0.05 |
| **Meta Training** | lr | 0.001 |
| | weight decay | 0.0005 |
| | milestone | "10" |
| | pruner reg | 10 |
| **Final Pruning** | metanetwork epoch | 38 |
| | speed up | 8.90 |
| Finetuning After Metanetwork | epoch | 2000 |
| | lr | 0.01 |
| | weight decay | 0.0005 |
| | milestone | "1850, 1950" |
| Finetuning After Pruning | epoch | 2000 |
| | lr | 0.01 |
| | weight decay | 0.0005 |
| | milestone | "1850, 1950" |

## D.4  RESNET50 ON IMAGENET

### D.4.1  EQUIVALENT CONVERSION BETWEEN NETWORK AND GRAPH

We change ResNet50 into a graph with node featrues of 8 dimensions and edge features of 1, 9 or 49 dimensions. We employ 3 linear layers to project edge features of these 3 different dimensions into the same hidden dimension, enabling their integration into our graph neural network.

**The node features** consist of 4 features derived from the batch normalization parameters (weight, bias, running mean, and running variance) of the previous layer, along with 4 features from the batch normalization of the previous residual connection. If no such residual connection exists, we assign default values $[1, 0, 0, 1]$ to the corresponding 4 features.

**The edge features** are constructed by simply flatten convolutional kernels into feature vectors (We have convolutional kernels of size $1 \times 1$, $3 \times 3$ or $7 \times 7$). We treat all residual connections as convolutional layers with kernel size $1 \times 1$. For example, consider two layers A and B connected via a residual connection, with neurons indexed as $1, 2, 3, \ldots$ in both layers. If the residual connection has no learnable parameters (i.e., it directly adds the input to the output), we represent the edge from $A_i$ to $B_i$ as a $1 \times 1$ convolutional kernel with value 1. Different from section D.2, to reduce the VRAM footprint of the program, we build no edges between $A_i$ and $B_j$ ($i \neq j$). If the residual connection includes parameters (e.g., a downsample with a convolutional layer and batch normalization), we construct the edge features in the same way as for standard convolutional layers.

### D.4.2  META TRAINING

**Data Models :** We generate 3 models as our data models, among them, 2 are used for meta-training and 1 are used for validation (visualize the metanetwork). When generating each data model, we finetune them based on the pretrained weights for 30 epochs with learning rate 0.01 and milestone "10", then pruning with speed up 1.2920x, followed by a finetuning for 60 epochs with learning rate 0.01 and milestone "30". Finally, we can get a network with accuracy around 76.1%.

**One Meta Training Epoch :** We train with `torch.nn.parallel.DistributedDataParallel` (pytorch data parallel) across 8 gpus. Unlike in Section D.2 and Section D.3, at the beginning of each epoch, we evenly distribute different data models across the GPUs. For example, since we have two data models, we load one on four GPUs and the other on the remaining four GPUs. At each iteration, we forward all eight models (replicated across the 8 GPUs) through the metanetwork to generate eight new models. We then feedforward a large batch of ImageNet data, evenly distributed through these eight new models, to compute the accuracy loss. We also use the parameters of the eight new networks to compute the sparsity loss. Then we backward the gradients from both two losses to update our metanetwork. Each training epoch consists of (ImageNet data num / big batch size) iterations, meaning that one full pass over the entire ImageNet dataset when computing the accuracy loss is considered as one epoch .

**Training:** We train our metanetwork with learning rate 0.01, milestone "2", weight decay 0.0005 and pruner reg 10. Finally we use metanetwork from epoch 12 as our final network for pruning.

### D.4.3  GPU USAGE

Meta-Training and feed forward through the metanetwork during pruning requires large VRAM and needs to be run on NVIDIA A100. All other training and finetuning can be run on NVIDIA RTX 4090. In practice, we use 8 gpus in parallel.

### D.4.4  FULL RESULTS

All results are summarized in Table 14, where we repeat the pruning process 3 times. It is important to note that due to the design of our algorithm and implementation, we cannot prune the network to exactly the same speed up across different runs. For instance, when targeting a $2.3\times$ speed up, the actual achieved speed up may be slightly larger, such as $2.31\times$, $2.32\times$, or $2.35\times$.

The aggregated statistical results are presented in Table 15, where each value is reported in the form of `mean(standard deviation)`.

### D.4.5 HYPERPARAMETERS

See Table 16.

Table 14: ResNet50 on ImageNet Full

| Method | Base Top-1(Top-5) | Pruned Top-1($\Delta$) | Pruned Top-5($\Delta$) | Pruned FLOPs |
|---|---|---|---|---|
| DCP (Zhuang et al., 2018) | 76.01%(92.93%) | 74.95%(-1.06%) | 92.32%(-0.61%) | 55.6% |
| CCP (Peng et al., 2019) | 76.15%(92.87%) | 75.21%(-0.94%) | 92.42%(-0.45%) | 54.1% |
| FPGM (He et al., 2019a) | 76.15%(92.87%) | 74.83%(-1.32%) | 92.32%(-0.55%) | 53.5% |
| ABCP (Lin et al., 2020b) | 76.01%(92.96%) | 73.86%(-2.15%) | 91.69%(-1.27%) | 54.3% |
| DMC (Gao et al., 2020) | 76.15%(92.87%) | 75.35%(-0.80%) | 92.49%(-0.38%) | 55.0% |
| Random (Li et al., 2022) | 76.15%(92.87%) | 75.13%(-1.02%) | 92.52%(-0.35%) | 51.0% |
| DepGraph (Fang et al., 2023) | 76.15%(-) | 75.83%(-0.32%) | - | 51.7% |
| ATO (Wu et al., 2024) | 76.13%(92.86%) | **76.59%**(+0.46%) | **93.24%**(+0.38%) | 55.2% |
| DTP (Li et al., 2023) | 76.13%(-) | 75.55%(-0.58%) | - | 56.7% |
| ours | 76.14%(93.11%) | 76.13%(-0.01%) | 92.78%(-0.33%) | **57.2%** |
| ours | 76.14%(93.11%) | **76.24%**(+0.20%) | **93.09%**(-0.02%) | **57.1%** |
| ours | 76.14%(93.11%) | 76.08%(-0.06%) | 92.93%(-0.18%) | 56.9% |

Table 15: ResNet50 on ImageNet Statistics

| Pruned FLOPs (Speed Up) | Pruned Top-1 | Pruned Top-5 |
|---|---|---|
| 57.1%(2.33×) | 76.15%(±0.07%) | 92.93%(±0.13%) |

Table 16: ResNet50 on ImageNet Hyperparameters

| Types | Name | Value |
|---|---|---|
| **Compute Resources** | GPU | NVIDIA A100 & NVIDIA RTX 4090 |
| | parallel | 8 |
| **Batch size** | small batch size | $32 \times 8$ |
| | big batch size | $32 \times 8 \times 200$ |
| **Prepare Data Models** | data model num | 3 ( 2 + 1 ) |
| Generate Data Model | epoch | 30 |
| | lr | 0.01 |
| | weight decay | 0.0001 |
| | milestone | "10" |
| Initial Pruning | speed up | 1.2920 |
| Finetuning | epoch | 60 |
| | lr | 0.01 |
| | weight decay | 0.0001 |
| | milestone | "30" |
| **Metanetwork** | num layer | 6 |
| | hiddim | 16 |
| | in node dim | 8 |
| | node res ratio | 0.002 |
| | edge res ratio | 0.002 |
| **Meta Training** | lr | 0.01 |
| | weight decay | 0.0005 |
| | milestone | "2" |
| | pruner reg | 10 |
| **Final Pruning** | metanetwork epoch | 12 |
| | speed up | 2.3095 |
| Finetuning After Metanetwork | epoch | 200 |
| | lr | 0.01 |
| | weight decay | 0.0001 |
| | milestone | "120, 160, 185" |
| Finetuning After Pruning | epoch | 200 |
| | lr | 0.01 |
| | weight decay | 0.0001 |
| | milestone | "120, 160, 185" |

# E  MORE ABOUT ABLATION STUDY

## E.1  FINETUNING AFTER METANETWORK

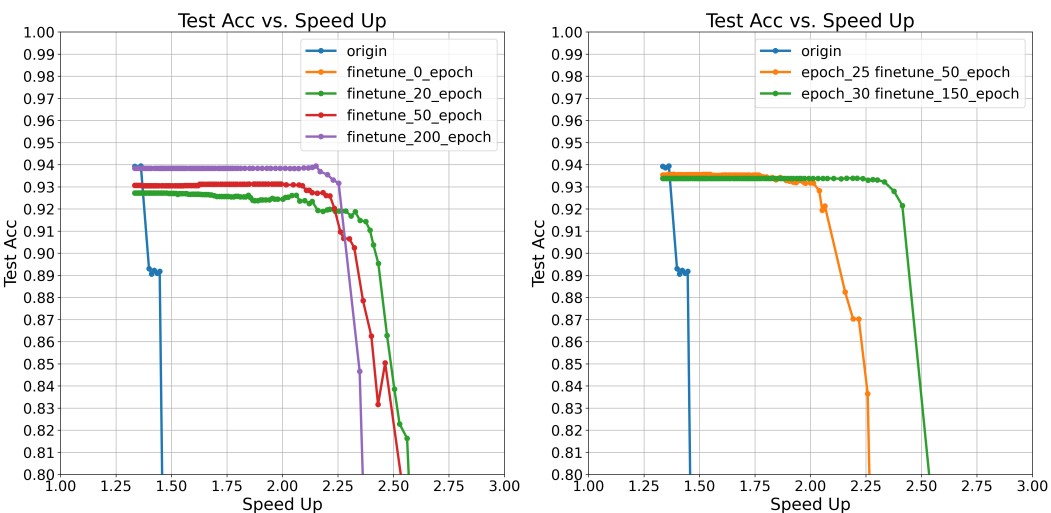

(a) Finetune Different Epochs with Same Metanetwork    (b) Use Finetune to Get Better Performance

Figure 11: Finetune After Metanetwork (ResNet56 on CIFAR10 as example)

As mentioned in section 5.1, our pruning pipeline can be summarized as:

$$\underbrace{\text{Initial Pruning (Finetuning)}}_{\text{Optional}} \rightarrow \underbrace{\text{Metanetwork (Finetuning)} \rightarrow \text{Pruning (Finetuning)}}_{\text{Necessary}} \qquad (29)$$

In this pipeline, finetuning after pruning is a common way to improve the accuracy. However, finetuning after the metanetwork is first proposed by us and may raise questions for readers regarding its impact on performance—whether it is necessary or beneficial ? In this section, we aim to clarify the effects of finetuning after the metanetwork and demonstrate that it is both necessary and advantageous.

To illustrate this, we use the example of pruning ResNet56 on CIFAR10. In Figure 11a, we visualize the "Acc vs. Speed Up" curves for models after the same metanetwork and after various amounts of finetuning. Without any finetuning after the metanetwork, the network's accuracy is nearly zero (which is why there is no "finetune_0_epoch" line in the figure—it lies at the bottom). Directly proceeding to pruning and finetuning from such an unfinetuned network experimentally results in poor performance.

However, as we increase the number of finetuning epochs after the metanetwork, several interesting observations emerge:

- **Quickly Recover:** The accuracy of our network after metanetwork quickly recovers after only very few finetuning, indicating that our metanetwork has the ability to preserve the accuracy (i.e. our accuracy loss in meta-training works)
- **Trend of Change:** As the number of finetuning epochs increases, the flat portion of the curve becomes shorter or stays the same, and the overall accuracy of this region improves.

When we feed the same network into metanetworks trained for different numbers of meta-training epochs and get several modified networks, then finetune them with different epochs to reach the same accuracy (as shown in Figure 11b), another important pattern emerges:

- **More Finetune Better Performance :** A network obtained from a metanetwork with more meta-training epochs can typically achieve the same accuracy as one from a metanetwork with fewer meta-training epochs—but requires more finetuning. Importantly, the former usually results in better pruning performance. This is evident in Figure 11b, where the green

line (representing a higher meta-training epoch) exhibits a longer flat region, indicating improved robustness to pruning.

This suggests that the effectiveness of our method can be further improved by employing metanetworks trained for more meta-training epochs and applying more finetuning afterward. In practice, however, we avoid excessive finetuning in order to keep the number of finetuning epochs within a reasonable range, allowing for multiple experimental trials.

Overall, all finetuning in our pipeline is necessary and effective.

## F EXPERIMENTS ON TRANSFERABILITY

### F.1 TRANSFER BETWEEN DATASETS

The common pruning pipeline is:

$$\text{Initial Pruning (Finetuning)} \rightarrow \text{Metanetwork (Finetuning)} \rightarrow \text{Pruning (Finetuning)} \quad (30)$$

Pruning pipeline for **None** is:

$$\text{Initial Pruning (Finetuning)} \rightarrow \text{Pruning (Finetuning)} \quad (31)$$

The dataset of a to be pruned network means we do all training and finetuning on it using this dataset. The dataset of a metanetwork means we generate dataset models, meta-train the metanetwork using this dataset.

We mainly have 2 hyperparameters here–*initial speed up* and *final speed up* . We approximately set them based on the difficulty of each dataset. Here dataset refers to the dataset of the to be pruned network (rows in Table 17). For CIFAR10, the initial pruning speed up is 1.32x and the final speed up is 3.0x. For CIFAR100 the initial pruning speed up is 1.32x and the final speed up is 2.5x. For SVHN, the initial pruning speed up is 3.0x and the final speed up is 10.0x.

Table 17: **Transfer between datasets**: All networks' architecture is ResNet56. Columns represent the training datasets for the metanetwork, and rows represent the training datasets for the to be pruned network. "None" indicates using no metanetwork. Results with metanetwork is obviously better than no metanetwork (The only exception is when training datasets for the to be pruned network is SVHN, and we guess this is because the dataset SVHN itself is too easy).

| Dataset\Metanetwork | CIFAR10 | CIFAR100 | SVHN | None |
|:---:|:---:|:---:|:---:|:---:|
| CIFAR10 | **93.35** | 92.47 | 92.87 | 91.28 |
| CIFAR100 | 69.97 | **70.16** | 69.25 | 68.91 |
| SVHN | 96.79 | 96.50 | **96.86** | 96.78 |

### F.2 TRANSFER BETWEEN ARCHITECTURES

The common pruning pipeline is:

$$\text{Initial Pruning (Finetuning)} \rightarrow \text{Metanetwork (Finetuning)} \rightarrow \text{Pruning (Finetuning)} \quad (32)$$

Pruning pipeline for **None** is:

$$\text{Initial Pruning (Finetuning)} \rightarrow \text{Pruning (Finetuning)} \quad (33)$$

The architecture of a to be pruned network means this network is constructed using this architecture. The architecture of a metanetwork means all dataset models we generated for meta-training this metanetwork use this architecture.

We mainly have 2 hyperparameters here–*initial speed up* and *final speed up*. We approximately set them based on the ability of each architecture. Here architecture refers to the architecture of the to be pruned network (rows in Table 18). For ResNet56, the initial pruning speed up is 1.32x and the final speed up is 3.0x. For ResNet110 the initial pruning speed up is 2.0x and the final speed up is 4.0x.

### F.3 TRANSFER FROM SMALL DATASET TO LARGE DATASET

When we are using large dataset like imagenet, is it possible that we use a much smaller dataset during meta-training but get the same results? Our answer is yes.

When pruning Resnet50 on IMAGENET, we evenly choose 10% on each class in IMAGENET and form a new subset dataset. Meta-Training using this subset causes almost no drop in accuracy. See results in Table 19. This suggest that even 10% percent dataset is enough in meta-train to let metanetwork learn the pruning strategy.

Table 18: **Transfer between architectures**. All training dataset is CIFAR10. Columns represent the architectures used for training the metanetwork, and rows represent the architecures of the to be pruned network. "None" indicates using no metanetwork. All results with metanetwork is obviously better than no metanetwork.

| Architecture\Metanetwork | ResNet56 | ResNet110 | None |
|:---:|:---:|:---:|:---:|
| ResNet56 | **93.40** | 92.81 | 92.08 |
| ResNet110 | 93.04 | **93.38** | 92.40 |

Table 19: During meta-train, use full ImageNet vs. only 10% ImageNet

| Method | Base Top-1(Top-5) | Pruned Top-1($\Delta$) | Pruned Top-5($\Delta$) | Pruned FLOPs |
|:---:|:---:|:---:|:---:|:---:|
| ours | 76.14%(93.11%) | 76.13%(-0.01%) | 92.78%(-0.33%) | 57.2% |
| ours(10%) | 76.14%(93.11%) | 76.24%(+0.10%) | 92.65%(-0.46%) | 57.0% |

### F.4 Transfer From Classification Task to Detection Task

We transfer the metanetwork from a classification task into another detection task. The detection dataset is PASCAL VOC 07 (Girshick et al., 2014). Our detection network is a Faster R-CNN detector with an ImageNet-pretrained ResNet-50 backbone (conv layers only, no FPN), a single-scale RPN with 5×3 anchors per location, RoIAlign to 7×7, and the default torchvision Fast R-CNN two-FC-layer head for 21-way VOC classification and bounding-box regression.

We use metanetwork traind when pruning Resnet50 on ImageNet. During pruning, we prune the resnet50 backbone with 2.5x speed up. Results are in Table 20. We can see that metanetwork does help preserve the detection ability of the network even if it is trained on a classification task. This is not a perfect experiment because nowadays there are much more different and stronger detection architectures and pretrain is widely used to enhance the network's ability. But adding too many complex structures would make the experimental results difficult to analyze. So we conduct this simple and fair experiment and it demonstrates that our metanetwork can transfer to different tasks.

Table 20: Transfer classification to detection

| Method | Origin | Prune w/o metanetwork | Prune with metanetwork |
|:---:|:---:|:---:|:---:|
| mAP | 0.6061 | 0.4524 | 0.5173 |

# G EXPRIMENTS ON TRANSFORMERS

## G.1 MHSA TO GRAPH

A MHSA (multi-head self-attention) begins with linear projections of the input $\mathbf{X}$ using $3H$ independent weight matrices—$\mathbf{H}$ each for queries, keys, and values. For each head $h \in 1, \ldots, H$:

$$Q_h = \mathbf{X}\mathbf{W}_h^Q, \quad K_h = \mathbf{X}\mathbf{W}_h^K, \quad V_h = \mathbf{X}\mathbf{W}_h^V.$$

Then each head computes attention via

$$Y_h = \mathrm{softmax}(Q_h K_h^\top)V_h,$$

The outputs $Y_1, \ldots, Y_H$ are concatenated and linearly projected:

$$\mathrm{MHSA}(\mathbf{X}) = \mathrm{Concat}(Y_1, \ldots, Y_H)\mathbf{W}^O.$$

The input and output are both $d$-dimensional. Each head produces $d_H$-dimensional outputs. When changing to a graph, we represent this with $d$ input nodes in the first layer, $H \cdot d_H$ attention head nodes in the second layer, and $d$ output nodes in the third layer.

Between the first and the second layer, we model the three projection types (query, key, value) using multidimensional edge features: for each edge $(i, j)$ in head $h$, the feature is

$$e_{ij}^h = \big((\mathbf{W}_h^Q)_{ij}, \ (\mathbf{W}_h^K)_{ij}, \ (\mathbf{W}_h^V)_{ij}\big).$$

Between the second and third layer, concatenation and the final projection $\mathbf{W}^O$ are handled naturally by the graph structure and treated as a standard linear layer.

In summary, every pair of nodes between the first and second layers is connected by an edge characterized by three-dimensional features, which represent the attention parameters. Each pair of nodes between the second and third layers is connected by an edge with a single-dimensional feature, corresponding to the final linear projection.

## G.2 EQUIVALENT CONVERSION BETWEEN NETWORK AND GRAPH

We change ViT-B/16 into a graph with node featrues of 6 dimensions and edge features of 1, 3 or 256 dimensions. We employ 3 linear layers to project edge features of these 3 different dimensions into the same hidden dimension, enabling their integration into our graph neural network.

**The node features** consist of 6 features, they are weight and bias of previous layer norm, bias of previous linear layer, biases of query, key, value of previous attention layer. In the case that any of these features doesn't exist, they are replaced by a default value of $[1, 0, 0, 0, 0, 0]$ at their corresponding positions.

**The edge features** are constructed almost in the same way as previous experiments. The only new part is MHSA, and we construct it as described in Appendix G.1.

## G.3 META TRAINING

**Data Models:** We use the default ViT-B/16 provided by PyTorch as the only data model. All training and pruning are conducted on this model, as training several separate models from scratch would be prohibitively time-consuming. Moreover, this choice ensures consistency with prior work in the field. Based on the Acc VS. speed up curve of the original ViT (Figure 12), we do initial pruning with a speed up of 1.0370x and apply no finetuning. This configuration yields a network with an accuracy of 81%.

**One Meta Training Epoch :** The same as ResNet50 on ImageNet (Appendix D.4.2).

**Training:** We train our metanetwork with learning rate 0.01, milestone "3", weight decay 0.0005 and pruner reg 1000. Finally we use metanetwork from epoch 6 as our final network for pruning.

Figure 12: Acc VS. Speed Up curve of the origin ViT

### G.4 GPU USAGE

Meta-Training and feed forward through the metanetwork during pruning requires large VRAM and needs to be run on NVIDIA A100. All other training and finetuning can be run on NVIDIA RTX 4090. In practice, we use 8 gpus in parallel.

### G.5 FULL RESULTS

All results are summarized in Table 21.

### G.6 HYPERPARAMETERS

See Table 22.

Table 21: ViT-B/16 on ImageNet

| Method | Base Top-1 | Pruned Top-1 | $\Delta$Acc | FLOPs |
|---|---|---|---|---|
| ViT-B/16 (Dosovitskiy et al., 2021) | 81.07% | - | - | 17.6 |
| CP-ViT (Song et al., 2022) | 77.91% | 77.36% | -0.55% | 11.7 |
| DepGraph (Fang et al., 2023) | 81.07% | 79.17% | -1.90% | 10.4 |
| MetaPruning(ours) | 81.07% | 78.26% | -2.81% | 10.3 |

Table 22: ViT-B/16 on ImageNet Hyperparameters

| Types | Name | Value |
|---|---|---|
| **Compute Resources** | GPU | NVIDIA A100 & NVIDIA RTX 4090 |
| | parallel | 8 |
| **Batch size** | small batch size | $128 \times 8$ |
| | big batch size | $32 \times 8 \times 200$ |
| **Prepare Data Models** | data model num | 1 |
| Generate Data Model | epoch | 0 |
| | lr | - |
| | weight decay | - |
| | milestone | - |
| Initial Pruning | speed up | 1.0370 |
| Finetuning | epoch | 0 |
| | lr | - |
| | weight decay | - |
| | milestone | - |
| **Metanetwork** | num layer | 3 |
| | hiddim | 4 |
| | in node dim | 6 |
| | node res ratio | 0.1 |
| | edge res ratio | 0.1 |
| **Meta Training** | lr | 0.001 |
| | weight decay | 0.0005 |
| | milestone | - |
| | pruner reg | 10 |
| **Final Pruning** | metanetwork epoch | 22 |
| | speed up | 1.7 |
| Finetuning After Metanetwork | epoch | 300 |
| | lr | 0.01 |
| | weight decay | 0.01 |
| | scheduler | cosineannealinglr |
| | label smoothing | 0.1 |
| | mixup alpha | 0.2 |
| | cutmix alpha | 0.1 |
| Finetuning After Pruning | epoch | 100 |
| | lr | 0.0001 |
| | weight decay | 0.0001 |
| | scheduler | cosineannealinglr |
| | label smoothing | 0.1 |
| | mixup alpha | 0.2 |
| | cutmix alpha | 0.1 |

# H   MORE VISUALIZATION OF STATISTICS

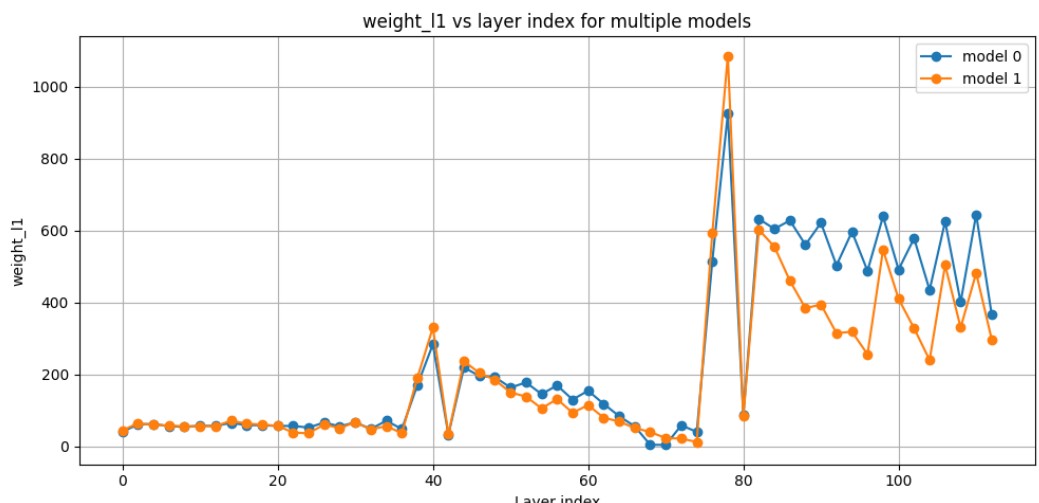

Figure 13: $l_1$ Norm

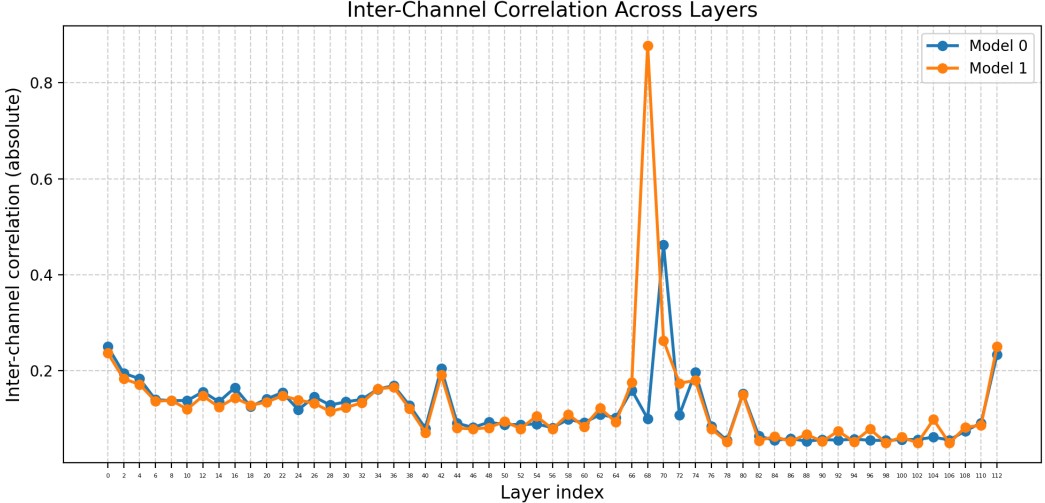

Figure 14: Inter-Channel Correlation

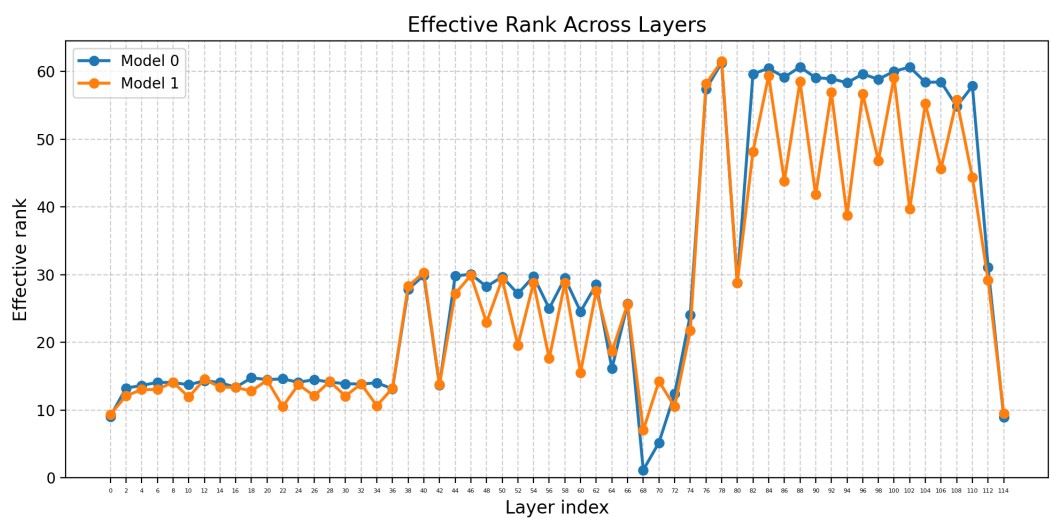

Figure 15: Effeicient Rank

# I   THE USE OF LARGE LANGUAGE MODELS

We only use LLMs to check grammar errors and polish our writing. All other works are done by human writers.

