# OpenReview forum: "Meta Pruning via Graph Metanetworks : A Universal Meta Learning Framework for Network Pruning"
_ICLR.cc/2026/Conference — Submitted to ICLR 2026_

### Official Review · Reviewer_WztN · 2025-10-25

**Soundness:** 4
**Presentation:** 2
**Contribution:** 3
**Rating:** 6
**Confidence:** 2

**Summary:**

The paper proposes a universal meta-learning framework for network pruning built around a graph metanetwork: instead of crafting new importance scores, a learned metanetwork takes a model that is hard to prune (under any chosen criterion like ℓ₁/ℓ₂) and transforms it into an easy-to-prune model; practically, the network is converted to a graph and passed through a GNN (PNA backbone) that outputs a modified network, after which standard pruning and finetuning suffice, and the trained metanetwork can transfer across datasets and architectures without per-task retraining—thus contributing (i) the introduction of metanetworks to pruning with a concrete graph-based implementation, (ii) a universal, criterion-agnostic pipeline requiring only feed-forward + finetuning, and (iii) empirical evidence of strong performance on ResNet56/CIFAR-10, VGG19/CIFAR-100, and ResNet50/ImageNet, alongside the finding that finetuning after the metanetwork is necessary to recover accuracy and improve robustness.

**Strengths:**

1. The paper reframes pruning via a metanetwork that takes a neural network as input and outputs a modified, easier-to-prune network—implemented as a graph metanetwork with a PNA-based GNN backbone—rather than inventing a new saliency score. It further argues broad, criterion-agnostic applicability and highlights transferability once a metanetwork is trained.

2. Strong empirical evidence across three classical CNN pruning tasks (ResNet56/CIFAR-10, VGG19/CIFAR-100, ResNet50/ImageNet) with headline improvements over most prior work, plus informative ablation via Acc-vs-Speed-Up curves showing slower accuracy decay after the metanetwork and finetuning. The paper also provides a concrete compute/memory cost comparison and amortization argument.

3. The pruning pipeline is explicitly specified and the paper is well-structured (clear sections for methods, ablations, transfer), includes a reproducibility statement, and discloses LLM usage limited to grammar polishing—improving transparency.

4. By offering a universal, criterion-agnostic route to make models easier to prune and demonstrating that a single trained metanetwork can be reused to prune many different networks with just feed-forward plus standard finetuning, the work has practical impact for reducing per-task engineering and scaling pruning to broader settings.

**Weaknesses:**

1. The metanetwork is implemented as a PNA-based message-passing GNN, i.e., an MPNN classically upper-bounded in expressivity by the 1-dimensional Weisfeiler–Leman (1-WL) test; thus non-isomorphic but 1-WL-indistinguishable architecture-graphs (e.g., $2C_{3}$ vs $C_{6}$) may collapse to the same representation, so the “optimal” meta-pruning policy could lie outside the realized function class.

2. The notion is presented qualitatively and illustrated ex post with accuracy–speedup plots, but it is not instantiated as a single, explicit training target. As a result, the training signals that steer the metanetwork may not be tightly aligned with the downstream pruning objective across architectures and sparsity regimes.

3. The method requires finetuning after the metanetwork and after pruning (thousands of epochs are listed in configs), and meta-training demands high VRAM. Actionable: provide wall-clock/energy accounting and amortization curves (meta-training once vs. number of downstream prunes), plus sensitivity analyses for meta-epochs and both finetuning stages.

4.  “Easy vs. hard to prune” remains descriptive rather than operational. The notion is presented qualitatively and illustrated ex post with accuracy–speedup plots, but it is not instantiated as a single, explicit training target. As a result, the training signals that steer the metanetwork may not be tightly aligned with the downstream pruning objective across architectures and sparsity regimes.

**Questions:**

Regarding the weaknesses of the paper, I raise the following questions：

1. Can authors demonstrate—on canonical 1-WL counterexamples constructed under your layer-DAG encoding(e.g., $2C_{3}$ vs $C_{6}$)—that the metanetwork yields distinct embeddings and different pruning decisions, or else provide a formal argument that such pairs cannot arise in your setting or would not alter the induced meta-pruning policy?

2. What single scalar objective is optimized during meta-training that aligns with the accuracy–speedup outcome (e.g., AUC over a fixed speed range or max speedup at a fixed accuracy floor), and what is its precise mathematical definition and cross-backbone correlation with the reported curves?

3. Can you provide a complete cost and amortization accounting—wall-clock, GPU-hours, kWh, and peak VRAM—for (a) meta-training, (b) post-metanetwork finetuning, and (c) post-pruning finetuning on a fixed hardware setup, together with the break-even reuse count versus strong baselines under matched accuracy/latency?

4. In the absence of an explicit scalar target, can you show that your current loss is a consistent surrogate for the area/shape of the accuracy–speedup curve across pruning budgets and architectures (e.g., via monotonicity, calibration, or rank-correlation plots)?

**Details Of Ethics Concerns:**

No Ethics Concerns

---

> ### Author Response · Authors · 2025-11-22
>
> We sincerely thank the reviewer for carefully reading our work and for providing many valuable questions and suggestions. We will address your weaknesses and questions as follows.
>
> ## Weakness 1
> >The metanetwork is implemented as a PNA-based message-passing GNN, i.e., an MPNN classically upper-bounded in expressivity by the 1-dimensional Weisfeiler–Leman (1-WL) test; thus non-isomorphic but 1-WL-indistinguishable architecture-graphs (e.g., \( 2C_3 \) vs \( C_6 \)) may collapse to the same representation, so the “optimal” meta-pruning policy could lie outside the realized function class.
>
> That's a really good question and you may be an expert in GNN theory. Yes, our GNN is bounded by 1-WL, and can't find "optimal" pruning policy that lie outside of it. But is this a weakness? We think the truth might not be the case. Please refer to our answer to Question 1 below.
>
> ## Weakness 2
> >The notion is presented qualitatively and illustrated ex post with accuracy–speedup plots, but it is not instantiated as a single, explicit training target. As a result, the training signals that steer the metanetwork may not be tightly aligned with the downstream pruning objective across architectures and sparsity regimes.
>
> Your insight on this problem is profoundly deep. And here are our thoughts. If you're asking there is any theoretical gurarantee that we can build a link between accuracy-speedup plots and the final pruning performance, our answer is no. But this is not only our problem, as far as we know, no pruning methods have such theoretical guarantee between pruning criterion and the final performance, because the network itself is too complex and processes like finetuning changes everything in an unpredictable way. However, if you're asking can we predict the pruning performance simply from accuracy-speedup plots emperically, our answer is yes. We do obserserve some emperical relationships and we add more details in "Relationship between "Acc VS. Speed Up Curve" and Pruning Performance" in Appendix D.1.1, you can refer to our revised paper for more details.
>
> ## Weakness 3
> >The method requires finetuning after the metanetwork and after pruning (thousands of epochs are listed in configs), and meta-training demands high VRAM. Actionable: provide wall-clock/energy accounting and amortization curves (meta-training once vs. number of downstream prunes), plus sensitivity analyses for meta-epochs and both finetuning stages.
>
> Please refer to our answer to Question 3 for wall-clock/energy accounting and amortization curves. All our experiments can be done with computation and memory costs in a resonable range. Sometimes in our experiments we finetune more epochs simply because this is the easiest way to persue higher performance and we don't want to tune hyperparameters (Many of our hyperparameters aren't carefully tuned, which suggesting our results can be better if we want, but that's not very meaningful so we didn't do it). Reduce the finetune epochs even by half won't cause much reduce in performance. Also, we already conducted ablation studies about finetuning epochs in Appendix E.1, you may find it useful.
>
> ## Weakness 4
> >“Easy vs. hard to prune” remains descriptive rather than operational. The notion is presented qualitatively and illustrated ex post with accuracy–speedup plots, but it is not instantiated as a single, explicit training target. As a result, the training signals that steer the metanetwork may not be tightly aligned with the downstream pruning objective across architectures and sparsity regimes.
>
> The same as our answer to weakness 2. We do have some "emperical training targets" that aligned with the downstream pruning objective and can use the accuracy-speedup plots emperically predict the pruning performance (Appendix D.1.1 Relationship between "Acc VS. Speed Up Curve" and Pruning Performance). However, there is no theoretical guarantee, as we mentioned in answer to weakness 2, and this is not only our problem, as far as we know, all pruning criteria don't have theoretical guarantee and just make sense intuitively and work practically.

---

> > ### Author Response · Authors · 2025-11-22
> >
> > ## Question 1
> > >Can authors demonstrate—on canonical 1-WL counterexamples constructed under your layer-DAG encoding(e.g., \( 2C_3 \) vs \( C_6 \))—that the metanetwork yields distinct embeddings and different pruning decisions, or else provide a formal argument that such pairs cannot arise in your setting or would not alter the induced meta-pruning policy?
> >
> > Our metanetwork(GNN) is bounded on 1-WL and can't distinguish 2C_3 and C_6, this do limit the theoretical expressivity of our metanetwork. However, we think this is not necessarily a problem, and even might be an advantege. The reasons are as follows:
> > (1) There is a gap between theory and practice. All our features come from network parameters, which can hardly be the same, so the C_3 or C_6 case is negligible.
> > (2) Higher expressivity often comes at the cost of hard to train and generalize. Though our GNN is only 1-WL, in a network a parameter has many linked parameters, which means every node in the graph already has many edges connected to it. Our experiments have shown that even do it in a 1-WL way can we reach very outstanding results and generalize well. If we change it to more complex architectures with stronger expressivity, it do offers more possibilities in theory, but in practice it becomes much hard to train and might not generalize as well as ours.
> > (3) Our GNN can be very small and saves VRAM use. We also tried some more complex GNN with stronger expressivity (2-WL) at first, but they generally require much more VRAM and are impossible to train.
> >
> > ## Question 2
> > >What single scalar objective is optimized during meta-training that aligns with the accuracy–speedup outcome (e.g., AUC over a fixed speed range or max speedup at a fixed accuracy floor), and what is its precise mathematical definition and cross-backbone correlation with the reported curves?
> >
> > We need to clarify 2 concepts:
> > (1) Meta-Training Object: According to our origin paper, Section 4.2, the only object in our meta-traning is two kinds of loss-accuracy loss and sparsity loss. So the meta-training object is to update metanetwork use the sum of these two kinds of loss.
> > (2) How to select the appropriate metanetwork: After meta-training we have many metanetworks from many epochs, and we want to choose the appropriate one for pruning. This time we visualize the "Acc VS. Speed Up" curve and evaluate it with something like max speedup at a fixed accuracy floor.
> > So we only optimze sparsity loss and accuracy loss during meta-training, and the trend of change in "Acc VS. Speed Up" curves of metanetworks at different epochs is a natural consequence. We find emperical relationship between the "Acc VS. Speed Up" curve and the pruning performance, so we use it to choose the appropriate metanetwork.
> >
> > The definition of "Acc VS. Speed Up" curve can be seen in Appendix C.1. The curves have almost same behavior across different backbones. In our origin paper main text Figure 5, we visualized "Acc VS. Speed Up" curves across various backbones and they share similar behavior.
> >
> > The definitions of meta-training object, the accuracy loss and sparsity loss are in our origin paper Section 4.3. It also points to the appendix with full details and mathematical definitions.
> >
> >
> > ## Question 3
> > >Can you provide a complete cost and amortization accounting—wall-clock, GPU-hours, kWh, and peak VRAM—for (a) meta-training, (b) post-metanetwork finetuning, and (c) post-pruning finetuning on a fixed hardware setup, together with the break-even reuse count versus strong baselines under matched accuracy/latency?
> >
> > Yes, that's a good idea. This will help readers further understand our method. We apologize for not doing this earlier. We've add relative contents in our revised paper, please refer to Appendix A.4.

---

> ### Author Response · Authors · 2025-11-22
>
> ## Quesiton 4
> >In the absence of an explicit scalar target, can you show that your current loss is a consistent surrogate for the area/shape of the accuracy–speedup curve across pruning budgets and architectures (e.g., via monotonicity, calibration, or rank-correlation plots)?
>
> Intuitively, our two losses in meta-training, sparsity loss and accuracy loss, aim at make the network more sparse while maintaining the accuracy. If a network is sparse, assuming an extreme case, half parameters are zero for example, then prune some parts of the network will cause no drop in accuracy. So if our meta-training losses works just as we expected, the network becomes more sparse, prune more of it causes no drop in accuracy and naturally leads to an Acc VS. Speed Up of a long flat region. We do observe this phenomenon, Acc VS. Speed Up curves across various architectures and datasets get a much longer flat region after metanetwork and fintuning, suggesting our losses may work as we expect. We also visualize some statistics in our revised paper, they showed more parameters have very small norm and lower importance (measured by taylor sensitivity) after metanetwork. Please refer to our paper Section 5.4 for more details, hope this can give you more inspirations.
>
> ---
>
> In all, we greatly thank you for proposing many valuable suggestions to our paper. It's so nice that you have raised many insightful thoughts. You inspired us to think more deeply and refine our paper. Thanks again, also hope our answer can give you some inspirations too.

---

> > ### Comment · Reviewer_WztN · 2025-11-26
> >
> > I appreciate the authors' feedback and detailed answers to my questions. I will maintain my current score.

---

> ### Author Response · Authors · 2025-11-27
>
> Dear reviewer,
>
> We thank you for your time and constructive feedback. Wish you every success in the future.
>
> Best wishes \
> Authors

---

### Official Review · Reviewer_2FBa · 2025-10-28

**Soundness:** 2
**Presentation:** 2
**Contribution:** 2
**Rating:** 2
**Confidence:** 4

**Summary:**

This paper proposes to train a meta-network to take as input a trained model and outputs a modified model that is easier to prune under a chosen criterion. Neural network pruning then proceeds with standard scoring and fine-tuning. The authors argue that existing pruning methods depend on fixed, hand-crafted criteria or require per-model learning-to-prune procedures with limited transferability. The proposed method aims to learn general rules that transform hard-to-prune networks into easy-to-prune ones. The idea is to represent the input network as a graph, run a GNN meta-network to predict small residuals on node and edge parameters, and convert the graph back to a network. The training uses task loss to retain accuracy and a sparsity-regularization loss which aligns with a structural-pruning criterion which is a group-norm score with corresponding sparsity loss. Experiments show that a single trained meta-network improves accuracy and speed-up curves on common computer vision tasks, and achieves SOTA trade-offs without model specific training. Additional experiments show flexibility to other pruning criteria and transferability across datasets and architectures.

**Strengths:**

The idea of using a meta-network to transform a neural network to a pruning friendly one is interesting and new to me. The details are properly executed to design the proposed method. Experimental results on three classical CNN pruning tasks and a ViT show good accuracy and speed-up curves. The proposed method is shown to be robust to pruning criterion and has good transferability across similar datasets and model architectures without re-training the meta-network.

**Weaknesses:**

The main experimental result table shows good numbers, but more baselines of recent structured, unstructured and hypernetwork pruning approaches would be desirable. The paper shows improved trade-off curves due to meta-network, but there’s limited analysis of which parameter and feature statistics, like channel saliency and inter-layer correlation, change. How much each model feature, like BN statistics, kernel encoding and residual edges, contributes to the performance. In general, I am not totally convinced that the proposed idea is always effective. It is also not clear if the conversion, padding and projection of edge features and the GNN’s memory and time footprint can scale to large backbones. How does meta-network depth and hidden layer size affect runtime and complexity. As the author claim that the framework works for any pruning criterion, experimental results on unstructured pruning and N:M sparsity are desirable. It is also not clear how to pick optimal meta-network checkpoint, as the meta-network training procedure does affect the accuracy and speed-up curve.

**Questions:**

1. After the metanetwork transformation, which statistics of the target model change in a way that explains the improved prune-ability? It is distribution of group-norm scores, inter-channel correlations, or sensitivity at each layer? Why does the proposed meta-learning method works generally?
2. How does the cost of network and graph conversion, and meta-network training and fine-tuning scale to larger pruning models and datasets?

---

> ### Author Response · Authors · 2025-11-22
>
> We sincerely thank the reviewer for carefully reading our work and for providing many valuable suggestions. Many of these comments are very helpful and will allow us to further refine our paper. We also note that some of the concerns you raised may stem from misunderstanding or missing certain parts of our paper, and we will provide further explanations below.
>
> # Weakness 1
> >The main experimental result table shows good numbers, but more baselines of recent structured, unstructured and hypernetwork pruning approaches would be desirable.
>
> We actually do include the relevant baselines in our paper — you may have missed them. For example, Table 1 presents a brief results, and its caption points readers to the full results with baselines in the Appendix: (1) Table 8, (2) Table 11, and (3) Table 14. Due to space limitations, we keep the main text focused on brief results and place the complete comparisons with prior work in the Appendix. Our baselines already cover nearly all related results from recent years, including structured, unstructured, and hypernetwork pruning methods.
>
> >The paper shows improved trade-off curves due to meta-network, but there’s limited analysis of which parameter and feature statistics, like channel saliency and inter-layer correlation, change. How much each model feature, like BN statistics, kernel encoding and residual edges, contributes to the performance.
>
> You're right and that's a great idea. We conduct further analysis on the statistics before and after metanetwork, please see our answer to Question 1.
>
> >In general, I am not totally convinced that the proposed idea is always effective. It is also not clear if the conversion, padding and projection of edge features and the GNN’s memory and time footprint can scale to large backbones. How does meta-network depth and hidden layer size affect runtime and complexity.
>
> In our paper Appendix A.2, we already give an estimation of computation and memory costs for scaling our work to larger models (also compared with prior classical methods). In short, our metanetwork doesn't need to scale with the to be pruned networks, so the general computation and memory cost is O(E^2), where E is the edge numbers of the to be pruned network. For more details, please refer to our origin paper. We have conducted experiments on relatively large models (resnet50 and ViT-B-16) and datasets (ImageNet). Our work may be not as inefficient as you think. In fact, it achieves outstanding performance while maintaining reasonable resource usage (faster than many existing complex pruning approaches).
>
> >  As the author claim that the framework works for any pruning criterion, experimental results on unstructured pruning and N:M sparsity are desirable.
>
> You are right. We sincerely apologize for previously only stating orally in the paper that our method is applicable to all kinds of pruning. We have conducted some additional experiments to show that our method also works for unstructured pruning and N:M sparsity pruning. Please refer to our revised paper, Section 6.2.
>
> >  It is also not clear how to pick optimal meta-network checkpoint, as the meta-network training procedure does affect the accuracy and speed-up curve.
>
> This can be seen in our origin paper, Appendix D, where we wrote all general experiment details. Especially in Appendix D.1.2 we wrote "Choose the Appropriate Metanetwork: During meta-training, we save the metanetwork after each epoch. After training is complete, we search for the most suitable metanetwork by visualizing its performance using a binary search strategy. Specifically, we start by visualizing the metanetwork from the middle epoch. If the accuracy of its flat region is below our target pruned accuracy, we next visualize the metanetwork from the first quarter of training. Otherwise, we check the one from the third quarter. We continue this process iteratively until we find a metanetwork that meets our criteria: a relatively high accuracy in the flat region and a long flat region indicating robustness to pruning." So the answer is already in the paper.

---

> > ### Author Response · Authors · 2025-11-22
> >
> > # Question 1
> > >After the metanetwork transformation, which statistics of the target model change in a way that explains the improved prune-ability? It is distribution of group-norm scores, inter-channel correlations, or sensitivity at each layer? Why does the proposed meta-learning method works generally?
> >
> > This is an interesting question ! We conduct some experiments and do find obvious changes in statistics. Please refer to our revised paper Section 5.4.
> >
> > # Question 2
> > >How does the cost of network and graph conversion, and meta-network training and fine-tuning scale to larger pruning models and datasets?
> >
> > In our origin paper Appendix A.2, we already estimate the computation and memory costs for scaling our work to larger models (also compared with prior classical methods). Please refer to our paper for more details. Also we add Appendix A.4 in our revised paper that give a concrete example of time, VRAM use and something like that compared with a representative prior work, you may also find it useful.
> >
> > ---
> >
> > In all, we greatly thank you for giving so many valuable suggestions to our paper. Especially for the statistics and unstructured pruning & N:M sparsity pruning, we've add relative experiments and add them to our paper. Our work has become more comprehensive and in-depth thanks to your suggestions. We also want to kindly clarify that you may miss some important parts in our paper and our work may not be as inefficient or weak as you think. Instead, our work has completely new idea, outstanding performance, sufficient experiments, great generability, is widely applicable and maybe more efficient than you may think.

---

### Official Review · Reviewer_sMiZ · 2025-10-29

**Soundness:** 3
**Presentation:** 3
**Contribution:** 2
**Rating:** 6
**Confidence:** 4

**Summary:**

In this work, the authors propose an network pruning method based on the meta-learning framework. The approach begins by representing the network to be pruned as a graph structure. Then, this method generates a new graph through meta network training (combining accuracy loss and sparsity loss) and returns it to the network.

**Strengths:**

The core advantage lies in its elimination of the need for specialized training for pruning each time. Training a meta network can prune any architecture network. Moreover, it also demonstrates strong generalization capabilities when transferring across datasets/architectures. Experimental details are fully described.

**Weaknesses:**

The main text often uses the expressions "no prior work has done something like this before" and "universally applicable". However, meta networks to predict network transformations for pruning is highly similar with the previous paradigm of meta learning pruning. In Numerical Experiments, the performance advantage is not significant enough. On ResNet56/CIFAR-10, there is no substantial difference in accuracy compared to DepGraph and ATO, and there is a certain improvement in “Speed Up”; On VGG19/CIFAR-100, the accuracy of proposed method (69.75%) is not as good as DepGraph (70.39%). The authors claim that it outperforms almost all work on three tasks, but overall it seems to be competitive rather than clearly leading.

**Questions:**

1.To more accurately reflect the contributions, the authors should clarify the difference between existing meta learning pruning and related work, based on Appendix A, and consider whether the expression is appropriate.

2.The paper conducted major experiments on ResNet56 on CIFAR10, VGG19 on CIFAR100, and ResNet50 on ImageNet. The experimental scope appears somewhat limited. It would be beneficial to evaluate the proposed method across a wider range of classic models on datasets such as CIFAR10, CIFAR100, ImageNet.

3.There are currently limited examples of "cross architecture/cross dataset". It would be meaningful if the author could provide more examples.

---

> ### Author Response · Authors · 2025-11-22
>
> We sincerely thank the reviewer for carefully reading our work and for providing many valuable questions and suggestions. We will address the weaknesses and questions as follows.
>
> ## Weakness 1
>
> >The main text often uses the expressions "no prior work has done something like this before" and "universally applicable". However, meta networks to predict network transformations for pruning is highly similar with the previous paradigm of meta learning pruning.
>
> No, we insist that our work is fundamentally new and no prior work has done something like this before. You may misunderstand our work or mix it up with some prior works. If you find any similar work, please tell us and we're willing to conduct further comparisons. If we do find someone has similar idea like us before we'll immediately quote their work in our paper and remove any claims like "no prior work has done something like this before" in our paper.
>
> >In Numerical Experiments, the performance advantage is not significant enough. On ResNet56/CIFAR-10, there is no substantial difference in accuracy compared to DepGraph and ATO, and there is a certain improvement in “Speed Up”; On VGG19/CIFAR-100, the accuracy of proposed method (69.75%) is not as good as DepGraph (70.39%). The authors claim that it outperforms almost all work on three tasks, but overall it seems to be competitive rather than clearly leading.
>
> Yes, you're right and we do admit our advantage is not significant. But pruning is already a well-studied field and the competition across all metrics is intense. Under this circumstances, we gather almost all recent years results among various tasks and use the same method get outstanding results in all tasks, it is by no means an easy task. Though DepGraph works better on VGG19/CIFAR100, but our method outperforms it on ResNet56/CIFAR10 and ResNet50/CIFAR100. So in all, you're right, our work isn't clearly leading, but it is really competitive and clearly leading all fixed pruning methods and learning to prune methods several years ago. Only very very few learning to prune methods in recent 2 years may can close to use or outperform us, but even outperform us, it is only on a limited subset of tasks while failing to match our results on others.
>
>
> ## Question 1
> >To more accurately reflect the contributions, the authors should clarify the difference between existing meta learning pruning and related work, based on Appendix A, and consider whether the expression is appropriate.
>
> Thank you for the suggestion. As noted, we have already clarified the distinction between existing learning-to-prune approaches and our proposed method in Appendix A. We believe the current presentation is appropriate. Our method is fundamentally new: to the best of our knowledge, it is the first meta-learning pruning methods using graph metanetworks and achieves outstanding results and great generality.
>
> In the revised paper, we expanded Section A.3 to better show our method's progress in generality and add Section A.4 to compare the memory and time cost with a prior work in a concrete example to help readers better understand our work. You can refer to the revised paper as you need.
>
>
> ## Question 2
> >The paper conducted major experiments on ResNet56 on CIFAR10, VGG19 on CIFAR100, and ResNet50 on ImageNet. The experimental scope appears somewhat limited. It would be beneficial to evaluate the proposed method across a wider range of classic models on datasets such as CIFAR10, CIFAR100, ImageNet.
>
> You may miss some important experiments in our paper. For traditional pruning, we conducted ResNet56 on CIFAR10, VGG19 on CIFAR100, ResNet50 on ImageNet, ViT-B-16 on ImageNet(Section 7, you may miss it). These experiments are the most popular and representative experiments that covers all mainstream architectures, from residual to no residual, from CNN to Transformer, from small dataset to large real dataset. While our method gets outstanding results on all these experiments, we think it's enough to demonstrate the effectiveness of our method.
>
> We believe that our experimental evaluation is already sufficiently comprehensive. Besides from the traditional pruning tasks, we also conduct a large amount of experiments on transferability, flexibility and wide applicability. After revised, we add some new experiments including unstructured pruning and N:M sparsity pruning, visualizing the statistics and adding more experiments on transferability. Please refer to our revised paper for these experiments, hope they can give you more inspirations.

---

> > ### Author Response · Authors · 2025-11-22
> >
> > ## Question 3
> > >There are currently limited examples of "cross architecture/cross dataset". It would be meaningful if the author could provide more examples.
> >
> > In our revised paper, we add Appendix F.3 and F.4, where we transfer from a small dataset (10% subset of ImageNet) to a large dataset (full ImageNet) and yields great results. Also we transfer our metanetwork which trained on classification task into detection task. For more details please refer to our revised paper.
> >
> > ---
> >
> > In all, we sincerely thank you for giving so many valuable suggestions to our paper. Your advice helps us refine our paper and improve its quality. Also, we want to kindly note that the our paper already presents a substantial number of experiments, which sufficiently validate the effectiveness of our approach. And our method is fundamentally new and nobody has done something like this before. Thanks again for your time and efforts.

---

### Official Review · Reviewer_sKJD · 2025-10-29

**Soundness:** 3
**Presentation:** 3
**Contribution:** 2
**Rating:** 4
**Confidence:** 4

**Summary:**

This paper proposes a meta-learning framework for network pruning, which can be applied to almost all types of networks.
The key idea is to use a metanetwork to transform a hard-to-prune network into an easy-to-prune one.
The framework consists of two phases: a meta-training phase, where a graph neural network learns transformation rules, and an application phase, where the trained metanetwork transforms a target network, followed by standard pruning and fine-tuning.
The authors claim that experiments on image classification tasks demonstrate that the proposed pruning framework achieves state-of-the-art performance.

**Strengths:**

- Introducing a metanetwork yields a reasonable accuracy improvement compared to the no-metanetwork baseline.
- Multiple ablation studies support the effectiveness of the proposed approach.
- Paper is well written and easy to follow

**Weaknesses:**

- Proposed metanetwork exhibits limited transferability across diverse model architectures and datasets.
Although the paper claims transferability as a key advantage (Section 4.4), the transfer experiments are restricted to highly similar settings: architecturally similar networks (ResNet56 vs. ResNet110) and datasets of similar scale and domain (CIFAR10/100/SVHN). To more convincingly demonstrate the claimed transferability, it could be better to include: (1) transfer across substantially different architecture families (e.g., metanetwork trained on ResNet applied to VGG or ViT without retraining), and (2) transfer across different dataset scales or vision tasks (e.g., a metanetwork trained on CIFAR applied to ImageNet-scale networks, or to other vision tasks such as detection).
- Potential requirement of many metanetworks to support various architectures and tasks.
- Limited performance improvement over existing methods on large-scale datasets (e.g., ImageNet).

**Questions:**

- How does the metanetwork perform on larger-scale datasets such as ImageNet or on architectures with substantially different scale and structure such as VGGs?
- Is it possible to build larger metanetwork that can support wide-range of tasks and datasets.
- Does Meta-Pruning also reduce the cost of fine-tuning after pruning? If the cost of fine-tuning remains comparable to dense fine-tuning, methods such as Soft-Threshold Weight Reparameterization [1], which jointly optimize weights and connectivity during fine-tuning, could potentially discover better weight–pruning pairs. Could the authors clarify whether the proposed fine-tuning process provides any concrete benefit in terms of efficiency or performance compared with such joint optimization approaches?

[1] Kusupati, A., et al. (2020). Soft threshold weight reparameterization for learnable sparsity.

---

> ### Author Response · Authors · 2025-11-22
>
> We sincerely thank the reviewer for carefully reading our work and for providing many valuable questions and suggestions. Your insightful comments help us better refine our paper. We also note that some of the concerns you raised may stem from misunderstanding or missing certain parts of our paper, and we will provide further explanations below.
>
> ## Weakness 1
> > Proposed metanetwork exhibits limited transferability across diverse model architectures and datasets. Although the paper claims transferability as a key advantage (Section 4.4), the transfer experiments are restricted to highly similar settings: architecturally similar networks (ResNet56 vs. ResNet110) and datasets of similar scale and domain (CIFAR10/100/SVHN).
>
>
> No, we think transferability is by no means our weakness but our strength. We agree with what you said, our transferability is limited across diverse model architectures and datasets. But our metanetwork can transfer between similar datasets and architectures and it is already a significant improvement compare to all prior learning-based pruning methods ! The case is the generality of learning-based pruning methods can be summarized as 4 stages from our point of view: (1) learning once prune one specific network. (2) learning once prune one type of networks(same architecture and dataset) as many as you want. (3) learning once prune one group of networks(similar architectures and datasets) as many as you want. (4) learning once prune any networks as many as you want. As far as we know, all previous learning to prune methods only reach stage (1), while our work robustly reach stage (3). So though our work doesn't reach stage (4) as you described, it is already a great success! We're sorry that we didn't make this clear in our origin paper, and we rewrite Appendix A.3 in our revised paper to further clarify our improvement in generality.
>
>
> > To more convincingly demonstrate the claimed transferability, it could be better to include: (1) transfer across substantially different architecture families (e.g., metanetwork trained on ResNet applied to VGG or ViT without retraining), and (2) transfer across different dataset scales or vision tasks (e.g., a metanetwork trained on CIFAR applied to ImageNet-scale networks, or to other vision tasks such as detection).
>
> For (1) we believe it is not possible for now, no one can do it. Because it requires at least build a super strong GNN and pretrain it on various architectures. That's a lot of work and we may consider take it as another new research in the future. For (2) we add some new experiments in our revised paper Appendix F.3 F.4. We found a metanetwork trained on CIFAR can't be applied to ImageNet-scale networks, but a metanetwork trained on a subset of ImageNet, using only 10% ImageNet in meta-train, can perform almost the same as using the whole imagenet. Also, we find metanetwork can transfer to detection tasks. Please refer to our revised paper for more details.
>
>
> ## Weakness 2
> >Potential requirement of many metanetworks to support various architectures and tasks.
>
> Yes, for very distinct tasks or architectures we need many metanetworks. But compared to all prior learning based pruning methods, they all need to retrain the learning process at each pruning process, which means one "learned model" for each network pruned, our work only require one "learned model" for a type of network pruned. Once the metanetwork is learned, we can use it to prune one type of network as many as we want, and even generalize to network with similar architectures or datasets. This is already much better than one "learned model" for each network pruned, and yields a great progress.
>
> ## Weakness 3
> >Limited performance improvement over existing methods on large-scale datasets (e.g., ImageNet).
>
> In prune ResNet50 on ImageNet experiment, our method still outperform almost all other works by a large margin. The only one close to us is "ATO" and it is a very new learning based pruning method. It is not that stable, requires much computational overhead, and aren't as generalizable as our method. (In fact, our methods are essentially different pruning ways, our is pruning after training, theirs are pruning while training).
>
> In summary, we have the confidence that our method achieves pretty outstanding performance. Unlike some prior work that may only compare with few selected prior work, our paper gathers almost all relative works in recent years and including various pruning ways, like fixed pruning and learning based pruning, pruning before or after training, etc.. Compared with all those works, our paper's outperforms almost all of them by a large margin (only with one or two expections sometimes), is already very impressive and by no means easy.

---

> ### Author Response · Authors · 2025-11-22
>
> ## Question 1
> >How does the metanetwork perform on larger-scale datasets such as ImageNet or on architectures with substantially different scale and structure such as VGGs?
>
> We have already conducted relavant experiments in our paper, you may miss it.
> See Section 5.2, Table 1, we report our results on pruning ResNet56 on CIFAR10, VGG19 on CIFAR100 and ResNet50 on ImageNet. The caption of the Table also indicates that full results (compared with prior works) on these experiments can be seen in Table 8, Table 11 and Table 14 in the Appendix. In Section 7 we also prune ViT-B-16 on ImageNet.
>
> So the answer is experiments in our origin paper already showed it performs well.
>
>
> ## Question 2
> >Is it possible to build larger metanetwork that can support wide-range of tasks and datasets.
>
> This is a very ambitious idea, and our answer can only be maybe. If you're asking simiar tasks and datasets, experiments in our paper already show it can. If you're asking a wide range of very different tasks and datasets, we think it is impossible and nobody can do it for now. Because this requires build very large metanetwork and pretrain it on various tasks, during this process, we may face difficulties like the VRAM is too large, the network learns one task and forgets another, or it can't generalize to new tasks etc.. It may be possible in the future if we scale everything up, but we're not sure and nobody is sure about this.
>
> ## Question 3
> >Does Meta-Pruning also reduce the cost of fine-tuning after pruning? If the cost of fine-tuning remains comparable to dense fine-tuning, methods such as Soft-Threshold Weight Reparameterization [1], which jointly optimize weights and connectivity during fine-tuning, could potentially discover better weight–pruning pairs. Could the authors clarify whether the proposed fine-tuning process provides any concrete benefit in terms of efficiency or performance compared with such joint optimization approaches?
>
> Meta-Pruning do reduce the cost of fine-tuning. Because our method performs better, only fewer finetuning is needed to achieve the same results as prior works—thus, from this perspective, we indeed reduce the amount of finetuning required. In our revised paper Appendix A.4, we proposed a concrete example that compare our work with a representative strong baseline in recent years and shows our methods require less time for finetuning when pruning after the metanetwork is trained. Coincidentally, the baseline we compare with in Appendix A.4 is DepGraph, whose idea is essentially very similar to [1], but it's a more recent work with clearer methods and wider applicability and better results. Please refer to our revised paper for full details.
>
> Compare to [1]:
> (1) As we mentioned above, our method do reduce the cost of finetuning, so your assumption "If the cost of fine-tuning remains comparable to dense fine-tuning" is wrong.
> (2) [1] can be only used in unstructured pruning, while our method can be used in both structured pruning and unstructured pruning.
> (3) Finetuning in [1] introduces extra parameters and training overhead, while our finetuning is pure finetuning with no other things and is much faster. In [1] paper, it says it finetune 100 epochs with 4 GPU needs around 3 days. While our finetuning ResNet50 on ImageNet needs 6 minutes per epoch with 8 NVIDIA RTX 3090, which means 10 hours 100 epoch, or 20 hours on 4 GPU, which is much faster.
> (4) [1] is a very old work in 2020, it do show great results at its times, but our work is the latest and outperform almost all previous works on various tasks, many of then are very strong baselines in recent years. So we have the confidence that our work is stronger than [1]. The way [1] uses soft-threshold helps make the network more sparse, but causes many biases in training processes and may cause large drop in accuracy compared to standard finetune.
>
> In all, Meta-Pruning do help reach higher performance with the same finetuning or reduce the cost of finetuning to reach the same results. [1] is also a great work with interesting ideas that once lead the development of pruning, but it do have some drawbacks compared to ours like introducing extra parameters and too much training overhead into finetuning, can only do unstructured pruning, and worce performance.

---

> ### Author Response · Authors · 2025-11-22
>
> In all, we greatly thank you for giving so many valuable suggestions to our paper. Especially for inspiring us revising some important parts in our paper, conducting more relative experiments and comparing with prior works to better understand our works. We also want to kindly clarify that you may miss some important parts in our paper and our work may not be as as weak as you think. Instead, our work has completely new idea, outstanding performance, sufficient experiments, and is widely applicable. Most importantly, transferability is definitely a strength of our paper. Though it seems limited as you mentioned in weakness 1, but it is already a huge improvement compare to all prior learning to prune works.

---

> > ### Comment · Reviewer_sKJD · 2025-11-27
> >
> > I appreciate the authors’ thoughtful feedback and detailed clarifications. The current experiments clearly show that the method performs well on similar datasets, but it is still unclear how far this similarity can be relaxed before the approach fails to generalize. Due to this open question, I will maintain my previous score.
> >
> > > We have already conducted relavant experiments in our paper, you may miss it. See Section 5.2, Table 1
> >
> > Thank you for pointing to Section 5.2 and Table 1. I did review those experiments, but my concern is that the authors don't evaluate how similar the architecture can the meta-network transfer. My question is whether the same metanetwork would still work for architectures that are more different, such as ResNet18 or ResNet34. This would help clarify how far the claimed architectural similarity actually extends.
> >
> > > Once the metanetwork is learned, we can use it to prune one type of network as many as we want, and even generalize to network with similar architectures or datasets.
> >
> > However, the paper does not seem to clarify how similar the datasets need to be for the metanetwork to generalize, nor how its performance changes as the dataset distribution becomes more different. Without systematically testing less related datasets, the claimed transferability remains under-validated and its practical scope is unclear. In addition, clarifying the range of dataset similarity would also help in evaluating whether the meta-learning cost is appropriately justified.

---

> ### Author Response · Authors · 2025-11-27
>
> Dear reviewer
>
> Thank you very much for the time and effort you have devoted to reviewing our work. Based on your comments, we kindly ask that you reconsider our submission. We are concerned that parts of our response may not have been fully understood by you, and we respectfully believe that our manuscript warrants a higher score.
>
> >The current experiments clearly show that the method performs well on similar datasets, but it is still unclear how far this similarity can be relaxed before the approach fails to generalize. Due to this open question, I will maintain my previous score.
>
> We respectfully clarify that this does not constitute sufficient grounds to reject our paper and assign a score of 4. You might think generalization problem is a weakness of our work. But the generalization ability of our method is already a significant improvement compare to all prior learning-based pruning methods ! And it is definitely our strength instead of our weakness. In our answer to weakness 1, we divided generalization ability of learning to prune methods into 4 stages: \
> (1) learning once prune one specific network. \
> (2) learning once prune one type of networks(same architecture and dataset) as many as you want. \
> (3) learning once prune one group of networks(similar architectures and datasets) as many as you want. \
> (4) learning once prune any networks as many as you want. \
> All previous learning to prune methods only reach stage (1), while our work robustly reach stage (3). It is unfair to reject our paper simply because "don't know when fails to generalize". Our work is not perfect, but it has already made substantial improvements compared to all prior works.
>
> About "how far this similarity can be relaxed before the approach fails to generalize", we've already conducted sufficient experiments showing it can generalize between similar datasets like CIFAR10, CIFAR100, SVHN and similar architectures like ResNet56 and ResNet110. It can also generalize from small dataset like 10% subset of ImageNet to full ImageNet, and from classification tasks to detection tasks. For networks of totally different architectures (CNNs to Transformers) or datasets from totally different backgrounds (languages to images), our work can't generalize of course and we believe nobody else can. As far as we know, this is absolutely enough to show the generalization of our method, no prior works can do it in a better way like create a universal math criterion and measure the accurate "generalization ability", that's impossible.
>
> >Thank you for pointing to Section 5.2 and Table 1. I did review those experiments, but my concern is that the authors don't evaluate how similar the architecture can the meta-network transfer. My question is whether the same metanetwork would still work for architectures that are more different, such as ResNet18 or ResNet34. This would help clarify how far the claimed architectural similarity actually extends.
>
> In our origin paper, we already conduct relative experiments in Section 6.3 Table 5 to show that our metanetwork can generalize between ResNet56 and ResNet110, so of course it can generalize between ResNet18 and ResNet34, we believe they are the same thing.
>
> >However, the paper does not seem to clarify how similar the datasets need to be for the metanetwork to generalize, nor how its performance changes as the dataset distribution becomes more different. Without systematically testing less related datasets, the claimed transferability remains under-validated and its practical scope is unclear. In addition, clarifying the range of dataset similarity would also help in evaluating whether the meta-learning cost is appropriately justified.
>
> As we mentioned before, we have already conducted many relavent experiments. If you want some criterion to measure the accurate "generalization ability", no prior works have done this before and we think nobody can because generalization the concept itself is only a machine learning theory and can't be accurately measured in almost all nowadays deep learning experiments. Also, generality is only one of the many strengths of our work, we have already conducted many relative experiments and we can't devote all our paper to generalization, we have many other strengths and we conduct experiments on almost all of them.
>
> ---
>
> In all, generality is definitely the strength of our paper rather than weakness. We also want to point out that generality is only one of the many strengths of our work, and it is not feasible for us to devote all the space in our paper to presenting it. You may misunderstand some parts of our paper and response. We believe our experiments are already sufficiently comprehensive. If you still have questions, we are willing to see any further discussion. However, we firmly believe that this does not constitute a valid reason to reject our paper and assign a score of 4.
>
> Best wishes
>
> Authors

---

### Author Response · Authors · 2025-11-22
**Paper Update**

Dear all:

We updated our paper ! The main updates are as follows:

1. Add experiments on unstructured pruning and N:M sparsity pruning. (Section 6.2)
2. Add ablation on statistics. (Section 5.4)
3. Appendix D.1.1 Add "Relationship between "Acc VS. Speed Up Curve" and Pruning Performance"
4. Add more compare with prior works on generality (Appendix A.3)
5. Add a concrete example compare with a representative prior work in time, VRAM and results. (Appendix A.4)
6. Add more experiments on transferability (Appendix F.3 transfer from small dataset to large dataset, Appendix F.4 transfer from classification to detection.)

Our work is a meta-learning based pruning methods that introduce graph metanetwork into pruning for the first time. It is fundamentally new and nobody has done something like this before. It is also a general framework that can be theoretically applied to almost all types of networks with all kinds of pruning and has great generality and transferability. We explains the framework step by step together with sufficiently comprehensive experiments demonstrating the effectiveness, flexibility and generality of our method. Many further ablation studies are conducted to gain a deeper insight into our method.

We greatly thanks the reviewers for your valuable suggestions that help us refine our paper. We appreciate the reviewers and ACs for your time and effort. We've answered your questions as much as we can and revised our paper based on your advices. If you have any further questions, please feel free to ask. We would be happy to continue the discussion.

Best wishes

Authors

---

### Author Response · Authors · 2025-12-01
**Final Response to AC (1)**

Dear AC,

This is our final response for you. We sincerely thank you and the reviewers for your time and effort. Due to some well-known reasons, the discussion phase has been stopped, so we write our final response here. We'll briefly introduce our paper and summarize the rebuttal phase to help you better understand our work.

## Brief Introduction of Our Paper
Our paper "Meta Pruning via Graph Metanetworks : A Universal Meta Learning Framework for Network Pruning" proposes a meta-learning framework for network pruning. It is a universal framework that can be applied to almost all types of networks with all kinds of pruning methods and achieves outstanding results on various tasks.

The framework can be summarized as: For a pruning criterion, we use a metanetwork to change a hard to prune network into another easy to prune network for better pruning. In the method section of our paper, we introduce
1. **How to build the metanetwork ?** We build a conversion between networks and graphs, and use a GNN(graph neural network) as our metanetwork.
2. **How to train the metanetwork ?** The meta-training process. We optimize the metanetwork using sparsity loss and accuracy loss, enable our metanetwork the ability to transform network easier to prune without disturbing the accuracy too much.
3. **How to design the pruning criterion ?** Most of our experiments are structural pruning and we build a serious of criteria based on group l2 norm (inspired from prior works). For  unstructured pruning, we simply use classic l1 norm.

Our work has several strengthes:
1. **Fundamentally new idea**: Our idea is fundamentally new and totally different from all prior learning to prune methods. We firstly introduce graph metanetwork into the field of pruning and let it learns how to "transform a hard to prune network into another easy to prune network". Once a metanetwork is trained, we can use it to prune as many networks as we want without any further training. All prior learning to prune methods need special training during each pruning and are fundamentally different from ours.
2. **Elegant and powerful architecture design**: Build a conversion between networks and graphs and use a GNN as our metanetwork elegantly captures the permutation equivariance of a network. The idea that GNN metanetwork uses information from neighbors to update local weights is very intuitive, performs well in practice, and naturally provides good generalization.
3. **Outstanding performance**: On various tasks (ResNet56 on CIFAR10, VGG19 on CIFAR100, ResNet50 on ImageNet, ViT-B-16 on ImageNet), our method outperforms almost all prior works. Especially, we gather almost all relative baselines, only 1 or 2 learning to prune methods in recent years reach results comparable to us, for all other baselines we outperform them by a large margin.
4. **Groundbreaking advance in generalization**: We divided generalization of learning to prune methods into 4 stages: (1). learning once prune one specific network. (2). learning once prune one type of networks(same architecture and dataset) as many as you want. (3). learning once prune one group of networks(similar architectures and datasets) as many as you want. (4). learning once prune any networks as many as you want.  \
As far as we know, all prior learning to prune methods only reach stage 1, while our work robustly reaches satge 3. Once a metanetwork is trained, we can not only use it to prune as many networks on the same architecture and dataset as we want, we can even use it to prune networks of similar architectures and datasets. No special training is needed during each pruning, only feed forward through metanetwork and standard finetuning. In our paper, all pruned network at test time are totally different from those that are used to train the metanetwork. Nobody has done something like this. For all prior works, they require special training during each pruning and can't generalize as ours.

5. **Wide applicability**: Theoretically our framework can be applied to almost all types of networks with all kinds of pruning. Experiments have shown it can be applied to almost all mainstream architectures--CNNs and Transformers, with and without residual connection, large and small models and datasets. It can be applied on almost all pruning types--sturctural pruning, unstructural pruning, N:M sparsity pruning. Very few pruning methods are so widely applicable.

6. **Well written methods and sufficient experiments:** Our paper has well written method in the main text with all relative techinical details and math formulars in the appendix. It also has plenty of experiments, some of them shows outstanding pruning results, some of them demonstrate the transferability and flexiblity of our method, and others are ablation studies to gain a deeper insight into out method. All strengths claimed in our paper are companied with relavant experiments.

---

> ### Author Response · Authors · 2025-12-01
> **Final Response to AC(2)**
>
> ## Rebuttal Summary
> ### sKJD
> He agrees that our paper has outstanding performance, supportive ablation study, and is well written and easy to follow.
>
> He also raises several weaknesses and questions. We managed to answer all of them. Some of them are misunderstanding or overlooking certain parts of our origin paper, so we refer to relative parts in our paper. Some of them are questions about our methods, we managed to answer all of them. He scores 4 and refuses to raise his score because he still thinks our method is weak in generality. While we've already clarified several times that generality is definitely our strength instead of our weaknesses. He may make this judgement due to the lack of background knowledge in learning to prune methods. And we insist our paper deserves a higher score.
>
> ### sMiZ
> He agrees that our work "elimination of the need for specialized training for pruning each time", has great generality and transferability , and fully described experiments.
>
> He questions about the writing of some parts in our paper and some of our experiments. None of them are big problems and we easily answer them all. He gives a score of 6 and doesn't show up in further discussion.
>
> ### 2FBA
> He agrees that our work has interesting idea, is robust and has good transferability.
>
> He gives a score of 2 and poses many weaknesses, while we find he might misunderstand or overlook many important parts of our paper and more than a half of his weaknesses and questions can find answer directly in our origin paper. We also add experiments on unstructural pruning and N:M sparsity pruning and some ablation studies on statistics according to his suggestions. Even we have pointed out all his misunderstanding or overlooking of our paper and finished all experiments he asked, he still refused to show up for further discussion and just remain a score of 2 even he doesn't understand our paper and missing many important parts in it.
>
> ### WztN
> He agrees our architecture is well designed, has wide generality and applicability, has outstanding performance. He also says our paper is excellently written, with comprehensive and well-structured content. And metanetwork can generalize and reuse is great.
>
> He gives a score of 6 and proposes questions more about some theories(including GNN expressivity and the meta-training object), we clearly answer them all. He also asks for a concrete example compare with prior baselines and report the VRAM and time usage, we add a relative experiment.
>
>
> ### Conclusion
>
> We managed to handle all problems proposed by reviewers and revised our paper based on their suggestions. WztN and sMiZ give score 6. sKJD insists to maintain score 4 because he thinks generality is our weakness even we've already clarified many times that generality and transferability is definitely our strengths rather than weaknesses. All other 3 reviewers mentions generality and transferatibility as our strengths too. So we think his reason to maintain the weak reject score isn't fair. 2FBa give a score 2. However, we think he misunderstands or overlooks many important parts of our origin paper, as more than half of his weaknesses and questions can find answers directly in our origin paper. We have finished all additional experiments he asked. but he refused to show up in the rebuttal phase and just remain a score of 2 even he doesn't understand our paper and missing many important parts in it.
>
> ---
>
> Again, we thanks all reviewers and ACs for your time and effort. Hope all of you have a wonderful day.
>
> Best wishes
>
> Authors

---

### Meta-Review · Area_Chair_9chD · 2025-12-04

**Summary:**

The main reviewer concerns are:

1. Limited transferability across different architectures and datasets
2. Overclaiming of performance

I don’t think these concerns have been dealt with during the discussion. The proposed method is competitive with baselines, but isn’t outstanding and there isn’t evidence that the meta-networks generalise across different architectures and datasets. The scores are 4 2 6 6, so this is borderline, and I propose rejection as I think the concerns are both quite serious.

**Reviewer Concerns:**

I think the most important concern (limited transferability) isn’t really addressed as comparisons are only given between different types of ResNets and standard image datasets. The authors claim in their response that it would be impossible instead of attempting experiments to demonstrate feasibility or otherwise, so I do not believe this is addressed.

Two reviewers note that results are competitive rather than leading. Indeed, the performance difference between the proposed approach and baselines outlined in the appendix tables is very minimal, which is at odds with the claim of “outstanding results” made by the authors.

**Reviewer Scores:**

Some reviewers have already stated what they planned to do before the discussion was frozen. Reviewer sKJD said they would retain their score of 4. Reviewer sMiZ gave a positive score (6) but wanted to see more examples of architecture/dataset transferability. The author has responded with a part of Imagenet to Imagenet experiment and applying a pruned network to a detection task as backbone. I don’t think the reviewer would have raised their score as these are very similar environments for transfer. I don’t think Reviewer 2FBa would have raised there score, although the authors have gone some way to answering their questions (although I don’t think at the level of depth that would be expected).

---

### Decision · Program_Chairs · 2026-01-26

Reject